# Wireless Electromagnetic Radiation Assessment Based on the Specific Absorption Rate (SAR): A Review Case Study

Mohamed Abdul-Al [1], Ahmed S. I. Amar [1,2], Issa Elfergani [1,3], Richard Littlehales [4], Naser Ojaroudi Parchin [5], Yasir Al-Yasir [1], Chan Hwang See [5,*], Dawei Zhou [6], Zuhairiah Zainal Abidin [7], Mohammad Alibakhshikenari [8], Chemseddine Zebiri [9], Fauzi Elmegri [1], Musa Abusitta [1], Atta Ullah [1], Fathi M. A. Abdussalam [1], Jonathan Rodriguez [1,3], Neil J. McEwan [1], James M. Noras [1,4], Russell Hodgetts [1,4] and Raed A. Abd-Alhameed [1,10,*]

1   Faculty of Engineering and Informatics, University of Bradford, Bradford BD7 1DP, UK; m.abdul-al@bradford.ac.uk (M.A.-A.); a.a.s.ismail@bradford.ac.uk (A.S.I.A.); i.t.e.elfergani@av.it.pt (I.E.); Y.I.A.Al-Yasir@bradford.ac.uk (Y.A.-Y.); felmegri@yahoo.co.uk (F.E.); mmabousitta@yahoo.co.uk (M.A.); a.ullah5@bradford.ac.uk (A.U.); fmakhzum@yahoo.com (F.M.A.A.); jonathan@av.it.pt (J.R.); njmcewan@cantab.net (N.J.M.); jmnoras3@gmail.com (J.M.N.); russell@sargard.com (R.H.)
2   Faculty of Engineering, Ain Shams University, Abassia, Cairo 11566, Egypt
3   Instituto de Telecomunicações, Campus Universitário de Santiago, 3810-193 Aveiro, Portugal
4   SARgard Ltd., Harrogate HG5 9DA, UK; richard@sargard.com
5   School of Engineering and the Built Environment, Edinburgh Napier University, Edinburgh EH10 5DT, UK; N.OjaroudiParchin@bradford.ac.uk
6   Honor Device Company Ltd., Shenzhen 518100, China; zhoudawei@hihonor.com
7   Faculty of Electrical & Electronic Engineering, Universiti Tun Hussein Onn Malaysia, Batu Pahat 86400, Malaysia; zuhairia@uthm.edu.my
8   Department of Signal Theory and Communications, Universidad Carlos III de Madrid, 28911 Leganés, Madrid, Spain; mohammad.alibakhshikenari@uc3m.es
9   Laboratoired'Electronique de Puissance et Commande Industrielle (LEPCI), Department of Electronics, University of Ferhat Abbas, Sétif -1-, Sétif 19000, Algeria; czebiri@univ-setif.dz
10  Information and Communication Engineering Department, Basrah University College of Science and Technology, Basrah 24001, Iraq
*   Correspondence: c.see@napier.ac.uk (C.H.S.); R.A.A.Abd@bradford.ac.uk (R.A.A.-A.); Tel.: +44-(0)-131-455-2683 (C.H.S.); +44-(0)-1274-23-4033 (R.A.A.-A.)

**Abstract:** Employing electromagnetic fields (EMFs) in new wireless communication and sensing technologies has substantially increased the level of human exposure to EMF waves. This paper presents a useful insight into the interaction of electromagnetic fields with biological media that is defined by the heat generation due to induced currents and dielectric loss. The specific absorption rate (SAR) defines the heating amount in a biological medium that is irradiated by an electromagnetic field value. The paper reviews the radio frequency hazards due to the SAR based on various safety standards and organisations, including a detailed investigation of previously published work in terms of modelling and measurements. It also summarises the most common techniques utilised between 1978 and 2021, in terms of the operational frequency spectrum, bandwidth, and SAR values.

**Keywords:** electromagnetic fields; EMF; SAR; radio frequency hazards

## 1. Introduction

Wireless Internet connectivity through a telecommunication network becomes an essential communication tool for machines [1], people [2], vehicles [3], infrastructures [4,5], and a wide range of devices that make up the Internet of Things (IoTs) [6,7]. This connectivity is realised using electromagnetic fields (EMFs) in the radiofrequency (RF) range and it shows exponential growth due to being implemented in new technologies, such as wireless wearable consumer electronics, personal health monitoring systems, mobile handsets, smart

meters, and household appliances. As a result, public concern has grown due to a substantial increase in the amount of human daily EMF exposure [8,9].

The characterisation of radiofrequency/microwave (RF/MW) fields from 3 kHz to 300 GHz and energy absorption have witnessed a tremendous advancement over the last five decades. This is due to the fact that researchers would like to understand and quantify the interaction mechanism between the biological entity and electromagnetic energy. Such biological effects and the risks they can carry have also been studied and revealed to be widely thermal [8,9]. It has been noted that the thermal health hazards issue was intensively considered in earlier investigations, while the later research tends to show increased concern for the effects of a human body on antenna performance. The meaning of the highest stated SAR (specific absorption rate) numbers for smartphones and many other wireless devices is often misunderstood. SAR is a measurement of the amount of RF energy absorption by the body from the source being measured, as shown in Figure 1, where SAR was evaluated when a smartphone was operated next to a human head [10].

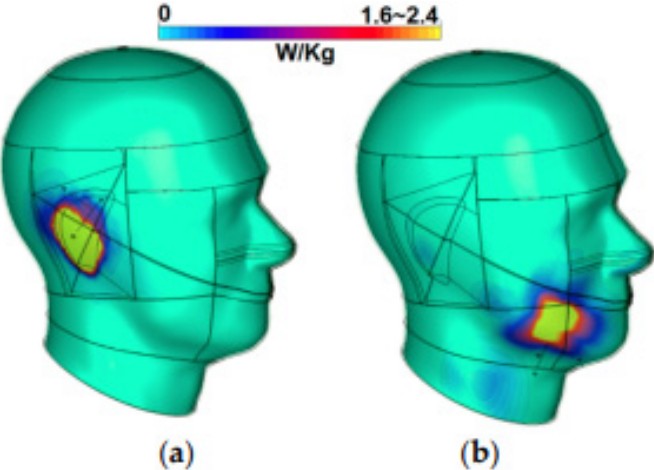

**Figure 1.** Analysis specific absorption rates of the smartphone antenna: (**a**) top location and (**b**) bottom location [10].

SAR is a simple method for assessing the RF exposure characteristics of smartphones to determine that they meet the FCC's (federal communications commission's) safety requirements [11]. SAR is often averaged across the entire body or over a limited part of the body; for instance, 1 g or 10 g of tissue. The specified value is again the maximum level measured in the bodily component under consideration over the given mass or volume. The evaluations of SAR methods will also be reviewed in this paper.

The current work is organised as follows. Radio frequency hazards are described in detail in Section 2. Section 3 defines SAR with mathematical equations. All the safety standards and organisations that are responsible for the requirements will be explained and outlined in Section 4. Section 5 will elucidate the numerical and experimental investigations of the SAR. This paper comes to an end in Section 6.

## 2. Radio Frequency Hazards

Radio frequency radiation is exposed over a larger scale of areas when it is produced by the radar, broadcasting, and communication services. However, when it is emitted by medical [12–14], industrial, as well as consumer devices [15], it only covers small areas. There has been a general trend in systems with more users and shorter ranges. Both the emissions of RFs and the reflection and scattering of electromagnetic waves by more than one source may cause the phenomena of a complex condition, namely multi-path propagation and spatially non-uniform fields. This has become a major limit to data rates, but is being circumvented by advances in signal processing.

Several investigations on how the human body can be severely affected by the exposure of RF electromagnetic fields approved that such a hazard should be dealt with. Here, the effect of the RF energy, and in particular when the standard level is exceeded, can cause numerous health issues [16,17]. These effects depend on many different factors, for instance, the modulation type used, the given frequency, the polarisation characteristics, as well as the distance from the source.

The calcifications of electromagnetic (EM) waves may be considered as both ionising radiations or non-ionising radiations according to their frequency and photon energy. Ionising radiations are extremely high frequency EM waves, such as those in gamma rays and X-rays. These waves produce sufficient photon energy to generate ionisation when the atomic bonds that hold molecules in a cell together are broken. Non-ionising radiation (NIR) contains photon energies that are not strong enough to break atomic bonds. This includes several forms of waves, namely, visible light, ultraviolet (UV) radiation, infrared radiation, microwave and radiofrequency fields, extremely low frequency (ELF) fields, as well as static electric and magnetic fields. The ionisation in a biological system cannot be produced due to the high intensity NIR. However, this does not mean that the NIR may not cause other biological effects. NIR creates some other biological effects in the process of heating, altering chemical reactions, or inducing electrical currents in tissues and cells [8,9,18], including their interactions with neuronal tissue [19,20].

These biological effects can be induced by electromagnetic waves and may lead to some harmful health issues. Thus, it is pivotal to distinguish between the two terms. The biological effect is normally generated by exposure to electromagnetic waves causing some physiological changes in the biological system [11,18]. Harmful health effects may occur when the biological effect is in the abnormal range of the body for compensation, which in turn may lead to certain unfavourable health circumstances. Since the thermal effects of non-ionising radiation on the cellular structure of live creatures are now well understood, the study of this topic was prioritised early on. For a thermal effect to be harmful, the tissue must be heated by at least 1 degree Celsius. The standardisation bodies use this as a foundation for their rules.

Engineers and scientists assess what the increased body temperature can be compensated for by transferring heat via the skin when determining the NIR threshold values. The body will no longer be able to cope with the additional heat if the temperature rises by more than 1 °C. Thermodynamic effects appear at this point. Reduced eye opacity, mental ability, and a variety of other physiological responses to heat are all well-known impacts of heat [21]. Worse effects, such as RF shock and burns, are established for very high temperature increases. Knowing about these side effects raises questions about whether mobile phone RF radiation is strong enough to heat bodily tissue. Although there have been no verified examples of people claiming to have had symptoms, such as headaches and dizziness as a result of using mobile phones, there are known cases of people claiming to have experienced symptoms, such as headaches and dizziness, as a result of using them [21].

Studies of the isolation of pig eyes that were exposed to various spectrums of pulsed microwaves [22] at 1 GHz, 3 GHz, and 35 GHz have been conducted by Hall et al. at Bristol University Hospital Department of Medicine. The position of the eyes was chosen to be in bony orbits of the human head model, and fluor optic probes were exploited to measure the temperature changes. An interesting finding was that not only was the corneal surface found to be hotter, but an increase of 5° occurred in the retinal temperatures. These investigations and tests exploited time-varying power; thus, as was expected, the cornea became hotter, the longer the exposure lasted. Therefore, issues, such as ocular discomfort and eye dryness, might be linked to microwave exposure. The duration of the testes varied from 6 to 12 min. In the same run of the tests, Hall [22] calculated that with exposure at 1.36 GHz, 5.3 GHz, and 9.6 GHz, the retinal temperatures increased by 5 °C, and 3.8 °C for 8.7 GHz at the same power level. Hall concludes that the increase in the temperature within the eye depended on various factors, namely incident power, microwave spectrum, the location in the eye, as well as the exposure duration.

Combining the frequency and incident power plays an important role in changing the pattern of corneal thickness. For example, the worst patterns can be seen at 1.36 GHz, 5.3 GHz, and 9.6 GHz, while the best ones occurred at 16 GHz and 34.8 GHz. Several different patterns of temperature increase within the eyes can be observed when different microwave frequencies are applied. Therefore, the existing problems in some types of mobile handsets may not be seen in other types that use different transmission frequencies. This agrees with the statement of Hall: "the position and size of thermal hotspots can lead to different physiological effects".

After David Reynard filed a lawsuit alleging that mobile phones caused his wife's death [23], the focus appears to have switched in the last several years. By November 1999, more than 200 researches on the health impacts of RF energy [23] were completed, and the focus switched to any long-term problems that might occur once the heat is no longer a concern. The so-called "non-thermal" effects have been observed in a variety of studies [24]. When an applied electromagnetic field generates no discernible heating impact, certain biological effects are noticed. The issue of non-thermal impacts is still a hot topic among researchers in the field. Many countries performed statistical research on the incidence of leukaemia among people living near low-frequency power transmission lines, with mixed results that contributed to a climate of public concern that tends to spill over into the radio realm. Even at power frequency fields of 50 Hz or 60 Hz, it appears that there is currently no way of knowing whether electric or magnetic fields are the likely cause of any problems.

There has been little clarification in the radio domain as to how many frequency regimes might contribute to any non-thermal impacts. Local microwave hyperthermy of tumours has been successfully adopted to enhance ionising radiation therapy, and RF diathermy has been widely employed by the medical profession for many years now with apparent success. Overexposure to microwave EM fields has also been linked to the development of cataracts in the eyes. This appears to be a thermal issue caused by a lack of blood supply to cool these highly specialised tissues.

Marha et al. [25] and Baranski and Czerski [26] presented several relevant investigations of microwave radiation's impacts. The stimulation of the neural system by electric fields is treated as a vital non-thermal action in these texts, but Schwan [27] refutes this. Refs. [28,29] showed that time-varying electric fields acting on nerve axons can cause sensory activation. The speculation that the fields of transceivers for cellular telephones may contribute to brain tumours is of current concern [30,31]. Anecdotal data suggests that those who operate with high-power HF (high frequency) transmitters have a higher risk of developing brain tumours. There was no doubt that the fields could have been extraordinarily strong in these instances. If the result is not thermal, there is no way to know what physical process might work in the vastly different frequency regimes of HF broadcast transmitters and personal communications systems. The possibility of a variety of different physical mechanisms, on the other hand, is even more difficult to visualise.

The complexity of the interaction between biological components and electromagnetic waves causes various challenges. Field strength may cause non-linear responses in cell structures. Some frequency windows or modulation methods may be more vulnerable to them than others. One question is whether continuous wave (CW) and pulsed wave (PW) radiation are different. The biological impacts of amplitude-modulated fields were demonstrated, including calcium ion mobility, changes in electrical brain activity, and intracellular alterations in enzymes [32]. Pulsed fields, on the other hand, are more likely to have biological effects since they create measurable effects at lower power levels than CWs [28]. When exposed to pulsed radiation, some people hear clicking sounds under certain circumstances. This may cause pain, even stress, as well as learning deficiencies [32]. The side-effects of the electrical stimulation of the brain have been illustrated and explained by A. T. Barker [33]. On the other hand, he mentioned that the harmful effects of brain simulation are hard to detect or identify, only occurring until a considerable time has passed. This aspect was also subsequently noted by Lezaks [34] and Groenwald. A decade later, Sienkiewicz of the National Radiological Protection Board wrote [35] that "Specific

ocular effects may also occur". He continued to confirm that various tissues of the eye can contained degenerative changes. This included both the lens and the retina following the protracted exposure to pulsed microwaves at certain levels, which were not carried in noteworthy heating effects. T. Barker stated that, whilst there is a kind of contradiction in terms of the data, serious effects have undoubtedly been found in some studies, whilst others have found the opposite [36].

As a result, the effects of NIR are not fully understood, and there is currently little evidence to deter individuals from spending too much time on their mobile devices. To date, there is only one fully established method of electromagnetic field interaction with biological material: heat generation due to induced currents and dielectric losses.

Since the early 1980s, researchers have been attempting to measure the amount of energy deposited in users of mobile communication devices. When it comes to a cell phone, this might be a difficult task. The incident fields in and around the head are complex functions of location since the phone is held so near to the user's head, making a straightforward analysis based on the plane wave incidence insufficient. Other issues are brought to light by the human head's complicated geometrical structure and variety. While mathematical solutions exist for the smallest geometries and highly simplified devices, their applicability to real-world models is severely limited. As a result, many field workers are employing a variety of numerical models to simulate the interaction of mobile phone handsets with the human body. How the electromagnetic field is transmitted and absorbed is determined by the electrical characteristics of the biological medium. Since the magnetic permeability of biological materials is effectively comparable to that of a vacuum, magnetic effects can be ignored. Biological materials' electrical characteristics are mostly determined by their water content, frequency, and temperature. The fraction of transmitted power absorbed by the skull for a mobile phone placed close to the operator's ear can exceed 40% [37], although the absorption is relatively localised and peaks close to the handset.

### 3. SAR (Specific Absorption Rate) Definitions

The electrical currents of RF in both the antenna itself as well as in the handheld case of the mobile device will certainly carry some RF electrical fields into the tissue. Consequently, some of this radiation energy may be absorbed by the human tissue, which leads to an increase in the tissue temperature. Such absorption phenomena are produced due to the power loss of dielectric polarisation. Water molecules' vibrations, movements of bound charges attached to macro-molecules, and movements of free ions, contribute most to the dielectric polarisation in biological material at radio frequencies.

Since the temperature changes generated by RF radiation cannot be directly measured, several specific characteristics are required to infer the degree of heating. The power density (W/m$^2$) indicates how much radiation passes through a certain surface and is thus appropriate for external exposure. It is common practice to define the SAR for internal exposure. SAR is a measurement of the amount of power absorbed per unit mass. The European Committee for Electrotechnical Standardisation (CENELEC) defines SAR as [38]: "The time derivative of the incremental energy (dW) absorbed by (dissipated in) an incremental mass (dm) contained in a volume element (dV) of given density ($\rho$)", or "The time derivative of the incremental energy (dW) absorbed by (dissipated in) an incremental mass (dm) contained in a volume element (dV)":

$$SAR = \frac{d}{dt}\left(\frac{dW}{dm}\right) = \frac{d}{dt}\left(\frac{dW}{\rho dV}\right) \tag{1}$$

SAR may be obtained by [39]

$$SAR = \frac{\sigma |E|^2}{\rho} \tag{2}$$

$$SAR = C_i \frac{dT}{dt} \tag{3}$$

$$SAR = \frac{J^2}{\rho\sigma} \tag{4}$$

$E$ = electrical field strength in the body tissue (V/m) and is defined by:

$$E = \sqrt{E_x{}^2 + E_y{}^2 + E_z{}^2} \tag{5}$$

where the $E_x$, $E_y$ and $E_z$, are the rms values of the $x$, $y$ and $z$ components of the electric field.

$\sigma$ = Body tissue conductivity (S/m);

$\rho$ = Body tissue mass-density (kg/m$^3$);

Ci = Body tissue heat capacity in J/kg °C;

$dT/dt$ = Body tissue time derivative of temperature in °C/s;

$J$ = Induced current density value in the body tissue in A/m$^2$.

At higher frequencies (>10 MHz), Equations (2) and (3) are commonly utilised. At lower frequencies, the induced current density, J, must be used to compute the worst-case exposure, which is commonly conducted in conjunction with Equation (4).

The definitions for three different parameters was reached: specific absorption, local peak SAR, whole body-average SAR. The parts of the body that are near the radiated device are influenced by the occurred absorption at frequencies above 300 MHz. Therefore, the local peak SAR limit is considered as the most critical value [40]. Practically, the SAR is defined as the average value in a finite tissue volume, while the power absorbed by the whole human body divided by the mass of the body is simply obtained by the whole-body average, SAR$_{Wba}$.

The basic constraints for the verification of mobile tele-communication equipment (EMT) compliance tests are specified in terms of the average SAR values over a tissue mass of 1 g (ANSI-IEEE (The American National Standards Institute-Institute of Electrical and Electronics Engineers) C95.1-1992, FCC) or 10 g (ICNIRP (International Commission on Non-Ionising Radiation Protection, April 1998)), and CENELEC (European Committee for Electrotechnical Standardisation) 50166-2). According to ICNIRP, a 1 g or 10 g tissue mass is defined as a cube that is "not a flat sheet on the surface" (ANSI-IEEE C95.11992, FCC CENELEC 50166-2) or as "any 10 g of continuous tissue" (April 1998). The transmission is pulsed in both circumstances, with pulses lasting less than 30 ps at frequencies greater than 300 MHz. A pulsed modulated field is an electromagnetic field created by one or more pulses amplitude modulating a continuous wave carrier [41]. Below are some details about the organisations that produce these standards.

## 4. Safety Standards and Organisations

Various national and international organisations have advocated safety recommendations based on the available knowledge. The following are some of the most important organisations:

1. The National Radiological Protection Board of the United Kingdom (NRPB). On both ionising and non-ionising radiation, the NRPB serves as a statutory adviser to the Health and Safety Commission and was one of the first to suggest a limit based on SAR [42].

2. In [43], the European Committee for Electrotechnical Standardisation (CENELEC) published detailed specifications for evaluating human exposure to electromagnetic fields.

3. The American National Standards Institute (ANSI) is a non-profit organisation. Until recently, the C95.1 committee was run by ANSI, but it is currently run by the American Institute of Electrical and Electronic Engineers (IEEE). The ANSI-IEEE document [44], which was first published in 1991 and modified in 1999, is the American standard for exposure limits. The most recent edition of this standard was published in 2019 [45].

4. The International Radiation Protection Association's (IRPA's) International Non-Ionising Radiation Committee (INIRC). This organisation has been in existence for quite some time. In May 1991, IRPA announced the formation of an International Commission on Non-ionising Radiation Protection (ICNIRP), whose objective is to investigate the risks posed by all types of non-ionising radiations. IRPA is a non-

governmental and non-political association of professional societies dealing with ionising and non-ionising radiation protection. In 1996, the ICNIRP released the "Guidelines for Limiting Exposure to Time-Varying Electric, Magnetic, and Electromagnetic Fields (up to 300 GHz)" [46], which was updated in 1998. In the year 2020, an updated version of this guidance was published.

5. The European Communities' Commission. The European Economic Community (EEC) Commission is currently drafting radiation protection rules, including the directive for the protection of workers against the risks of physical agents exposure and the machine safety directive. Radiation includes both ionising and non-ionising radiation [47].

6. The World Health Organisation (WHO). CENELEC and ANSI/IEEE [41] have both established standards for workers and the general public, commonly known as regulated and uncontrolled environments. In general, controlled settings are places where people are exposed to their knowledge (workers, employed people), while the uncontrolled environments are places in which the exposure caused to individuals occurs without their knowledge or control (general public).

The distinction is made because members of the general public, such as children and the elderly, may be affected at lower levels. Only those in good health are permitted in restricted environments. The limit for uncontrolled groups is set at ten, while the limit for managed groups is set at five. Table 1 summarises the SAR limits for the four most important safety standards.

**Table 1.** Summaries of the SAR limits for four main safety standards.

| References | Descriptions |
|:---:|:---:|
| [42] | ANSI/IEEE: averaged across any 1 g in the skull over any 6-min interval, 1.6 W/kg and 8 W/kg (for uncontrolled and controlled environments, respectively). |
| [44,45] | ANSI/IEEE: averaged throughout the entire body over an adequate average duration, 0.08 W/kg, and 0.4 W/kg (for both uncontrolled and controlled situations). |
| [43] | CENELEC: averaged over any 1 g in the skull over any 6-min interval, 2 W/kg and 10 W/kg (for uncontrolled and controlled environments, respectively). |
| [41] | CENELEC: averaged throughout the entire body over an adequate average duration, 0.08 W/kg, and 0.4 W/kg (for both uncontrolled and controlled situations). |
| [42] | NRPB: across any 6-min interval, 10 W/kg averaged over any 10 g in the cranium. |
| [42] | NRPB: throughout 15 min, 0.4 W/kg averaged over the entire body. |
| [46] | ICNIRP: 2 W/kg averaged over any 10 g in the skull for 6 min (for the general population). |
| [46] | ICNIRP: throughout 6 min, 0.08 W/kg averaged over the entire body (for the general population). |

## 5. Investigations of the Specific Absorption Rate

The first studies of power deposition in users at frequencies of about 900 MHz were mostly experimental. Balzano and his colleagues [48] employed a real human skull filled with a substance with conductivity and permittivity similar to brain tissue in a groundbreaking 1978 study. Support was supplied in the form of a rough apposition to the neck and shoulders. The phantom was illuminated using a 6 W radio tuned to 840 MHz.

The 2 antennas used were an 8 cm quarter-wave resonant whip stimulated against the set case at its base, and a 15 cm sleeve dipole isolated from the case by a choke and aroused in its centre. Several fascinating physical effects were detected, but they were barely mentioned in the more complex computational work that followed. The largest SARs are observed near the feed points of both antennas, where the fields are described as "of the low impedance", implying a small ratio of E to H field magnitudes. Electric fields in these areas are mostly tangential to both the antenna and the phantom surface, and they appear to induce high energy deposition that extends to larger depths in the material. The fields are high impedance at the antenna tips, and the electric fields are virtually normal

for the phantom's surface. These fields appear to be linked to a low SAR value, which is also concentrated in the fat layer's surface layers. The short whip antenna has nearly double the peak SAR of the sleeve antenna; however, the sleeve antenna has higher power densities in the vicinity of its feed point. The large current density in the case of the set where the whip antenna is used is interpreted in the present paper. This data suggests that magnetic field intensities, rather than electric field intensities, are the most critical factors in determining SAR.

Chatterjee et al. [49] provided direct measurements of SAR in head and body phantoms at numerous frequencies of up to 800 MHz, using realistic handset constructions, and found a peak value of roughly $1.3\,\text{kg}^{-1}$ at 800 MHz when the set was in contact with the phantom's chest, as shown in Figure 2. They explained that the overall trend of peak SAR increasing with the frequency is due to three factors: smaller resident antennas concentrate energy more, energy is absorbed more quickly, and less of the incident power is reflected.

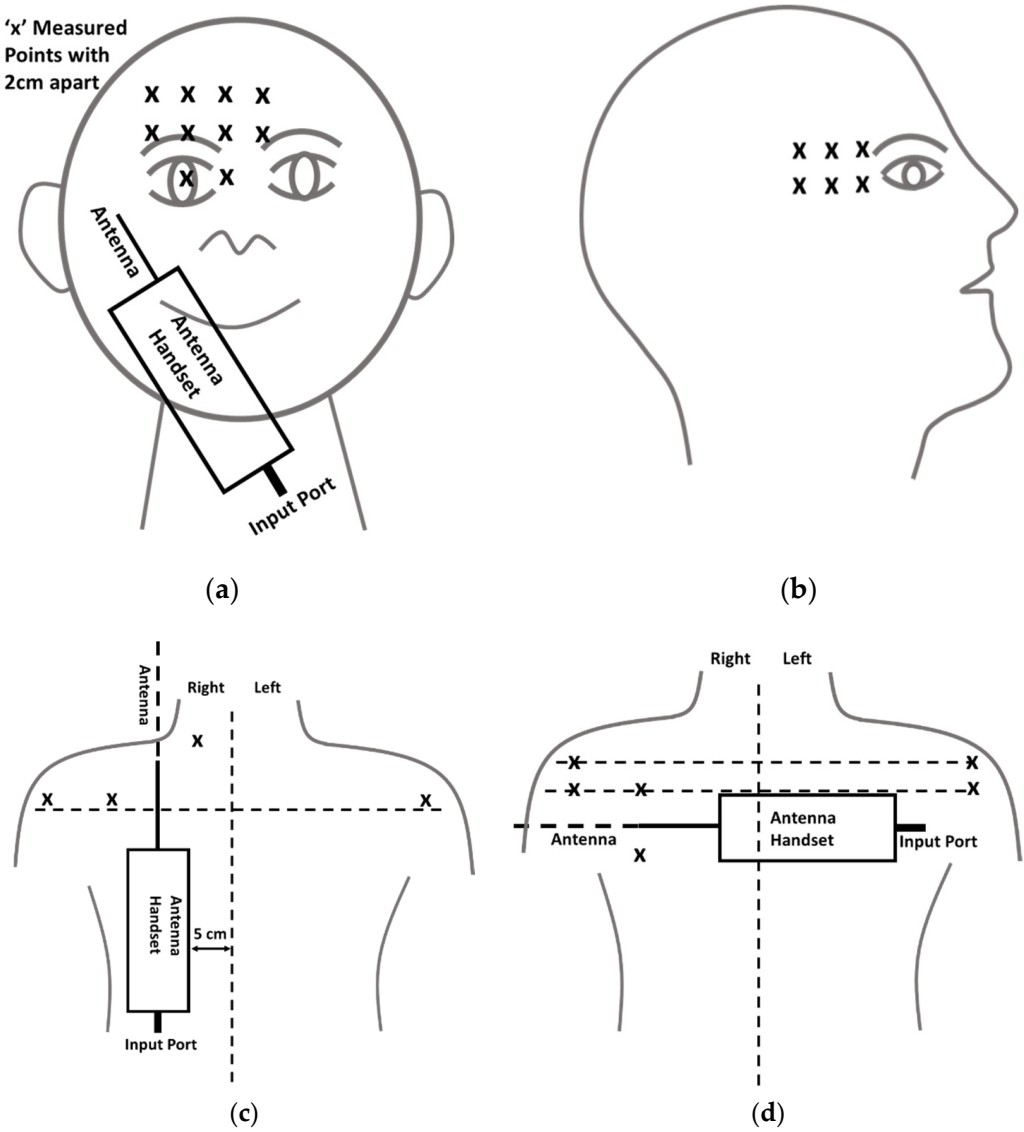

**Figure 2.** Direct measurements of SAR in the head and body phantoms: (**a**) front view—head measurement points (hand-held position, at an angle in front of the face with the antenna pointing upward and outward to the right), (**b**) head measurement points—side view, (**c**) body probe insertion points for position I, and (**d**) body probe insertion points for Position II.

In 1986, Stuchly et al. [50] offered a comparison of theoretical work and measurement data for human exposure in the near-field scenario of a type of dipole antenna at 350 MHz. For the electromagnetic computation, a model of the human body was used, as well as a tensor integral approach and the technique of moments.

A relatively early (1988) article by Sullivan et al. [51] is of interest as one of the first to use FDTD (finite-difference time-domain) modelling for the problem, but similar to [52], it only evaluated a maximum frequency of 350 MHz; therefore, both papers have little relevance for future substantial developments in mobile phone systems.

In 1990, Fujiwara et al. [53] used a closed-form formula for a homogeneous sphere irradiated by a half-wavelength dipole antenna to perform a calculation. With the emergence of anatomically precise models of the human head–body system around 1991, there have been significant improvements to the numerical computation of SAR.

Dimbylow [54] employed a plane wave to irradiate a realistic head model. They discovered some evidence of "hot spots" created by standing wave field structures and focussed on the the curved bones of the skull using FDTD to compute the fields in the head.

In 1993, Dimbylow [55] proposed results that were comparable to the commonly used head model, but this time it was irrigated by a dipole that was placed close to the head at 2 different spectrums: 900 MHz and 1.8 GHz. The absorption in the eye was computed as a function of the dipole's distance from the eye's surface in this situation. In this example, the head did not show a complex standing wave pattern, but rather a very basic SAR distribution that sharply peaked near the dipole and rapidly decayed with increasing distance within the head (particularly at 1.8 GHz), as represented in Figure 3. The dipole was not mounted on a genuine handset, and it was placed close to an eye that was designed to depict a worst-case scenario rather than a typical operational situation. When the dipole is 1.3 cm from the head, the estimated peak SAR at 1.8 GHz might reach a very high value of 22.9 kg$^{-1}$.

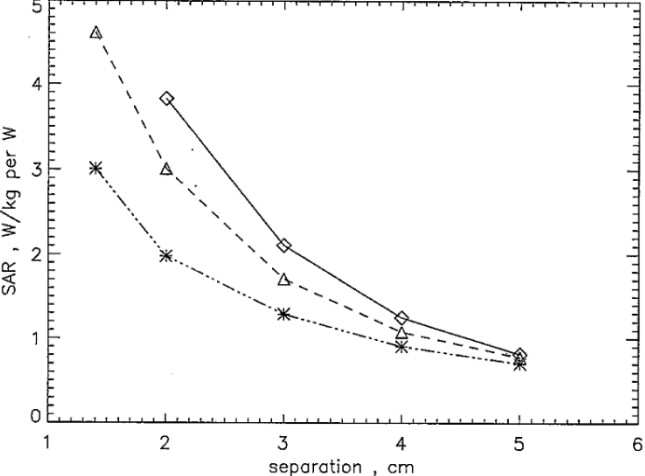

**Figure 3.** The envelope of maximum SAR averaged over 10 g at 1.8 GHz as a function of the separation of the centre of the transceiver from the head: AP$_v$ (diamonds), LAT$_v$ (triangles), and LAT$_h$ (asterisks).

In 1993, Kuster [56] used the multiple multipole (MMP) method to determine the fields. They employed a 6 cm helical dipole 2.5 cm from the head, which was a homogenous simplified non-spherical model, in addition to a skull layer. Although the 0.77 S/m value of conductivity assumed for the brain at 900 MHz appeared as minor compared to the figure used in [37], it is instructive to notice the peak value of 7.5 kg$^{-1}$ for the peak SAR in the model head.

The SAR level of the antenna–tissue interaction was evaluated by Jensen and Rahmat-Samii [57,58] from 1994 and 1995 at 915 MHz. The FDTD-based electromagnetic simulation along with actual antennas on the handset transceiver was setup. The planar inverted F antennas (PIFAs) were chosen to be a good example of both internal and external configu-

rations, as demonstrated in Figure 4. It was found that the peak and average SARs within the human head can be reduced when the PIFA internal antenna is placed on the back of the mobile device to be away from the user. Significantly, more details of those papers will be discussed later.

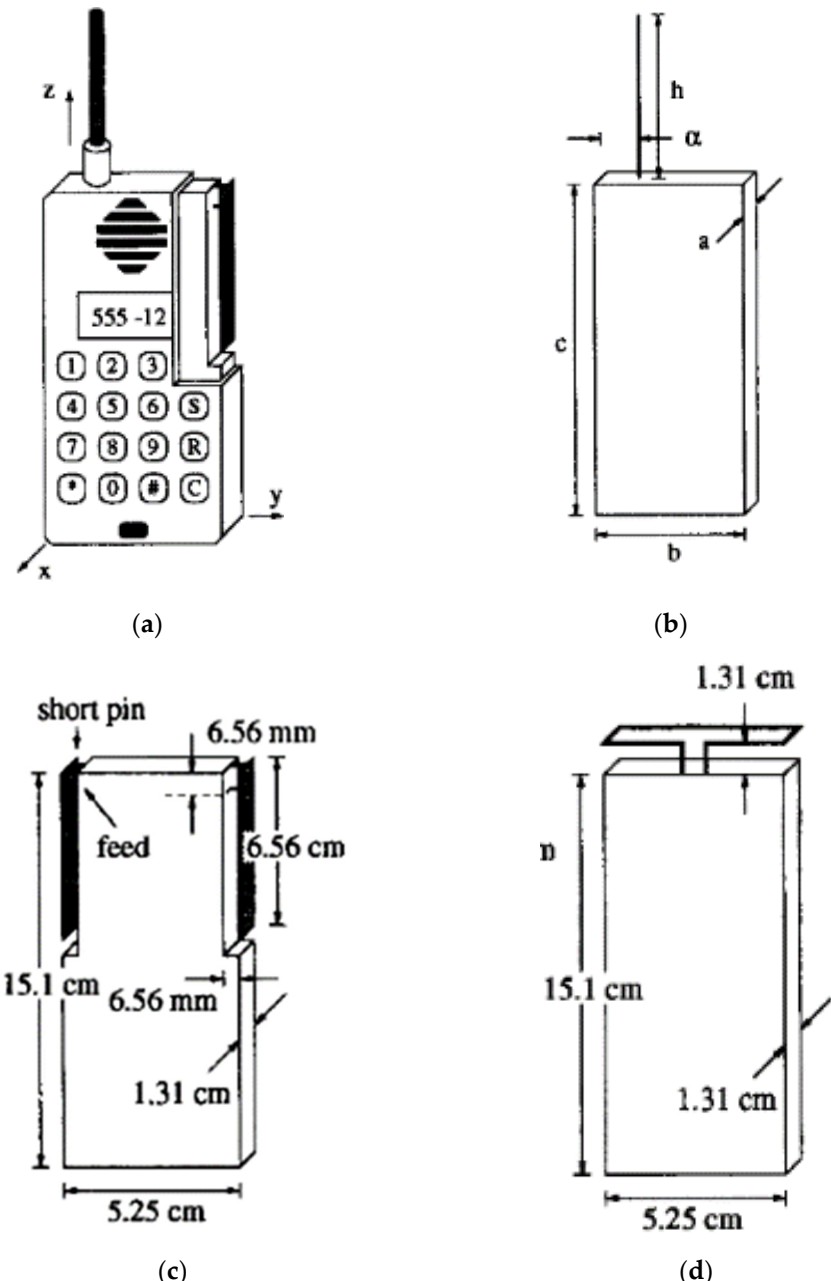

**Figure 4.** (**a**) Typical handset geometry showing the monopole and PIFA elements. Configuration and dimensions for the (**b**) monopole, (**c**) dual PIFA, and (**d**) top-mounted loop on the handset [55].

In 1996, Watanabe et al. [59] investigated the SAR in the human head models due to various exposure conditions: antennas of varying length from $\lambda/4$ to $\lambda/2$ mounted on top of metal handsets were modelled and placed close to the human head. The $\lambda/4$ antennas were excited against the handset, but the $\lambda/2$ antennas were excited at their centres and an air gap was placed between the bottom of the antenna and the handset. Frequencies of 900 MHz and 1.5 GHz were considered. Using the FDTD method they established that the maximum local SAR increased as the antenna length decreased. They also considered

models of the head, which included the auricle (visible part of the ear), and reported an increase in the maximum local SAR when the auricle was included in the model.

Furthermore, they stated that this effect was more pronounced for the longer antennas. The comparisons between computed SAR in homogeneous and heterogeneous models [60] of the head showed that, for the shorter antennas near the head, the difference in the maximum local SAR between the two models was quite small.

Finally, they included the hand wrapped around the metal handset and concluded that the position of the hand did not affect the maximum local SAR, providing it did not obscure the antenna. The peak SARs were much higher at 1.5 GHz than at 900 MHz. The difference in peak SAR between the $\lambda/4$ and $\lambda/2$ cases was much more pronounced for the 900 MHz than for the 1.5 GHz antennas. This can presumably be explained simply by the observation that the longer antenna at the lower frequency means that the displacement of the current peak away from the head is greater. At both frequencies and for both antenna types, the peak SAR was observed to vary roughly as $d^{-3/2}$, where d is the distance between the axis of the antenna and the auricle.

Gandhi et al. [37] used the FDTD method to study the SAR induced in the head with variations in the antenna, head model, and tissue properties at frequencies 835 MHz and 1900 MHz. Monopole antennas of lengths $\lambda/4$ and $3\lambda/8$ were used in both the vertical position and tilted 30° vertically. The longer antennas resulted in lower SARs. The rotation of the handset lowered the SAR at the lower frequency, but had little effect at 1900 Mhz. Using the handset dimensions previously used by Dimbylow [55], a different head model was used and evaluated.

The l g SAR agreed within 10% with those of Dimbylow. Of note is the reduced-size head models employed to represent children of l0 and 5 years old. In these smaller models, larger SAR values were reported, as was a greater in-depth penetration, especially at 835 MHz. Additionally, the effect of using a truncated head model was considered, in which 1/2 or 2/3 of the head from the non-handset side was removed. Truncating the head models did not significantly alter the SAR distribution in the head. Interestingly it was reported that the use of a homogenous head model overestimated the maximum l g SAR by approximately 42%.

In 1996, Hombach et al. [40] examined the energy absorption dependency of the human head anatomy and the models used to approximate it at 900 MHz. Up to 13 tissue types with varied electrical characteristics were detected using various numerical head "phantoms" based on the MRI (magnetic resonance image) scans of 3 people. In a multi-tissue phantom and two homogeneous phantoms of various forms and sizes, computed results were compared to the measured values. They observed three different "numerical" phantoms based on MRI scans that revealed up to 13 different tissue types, and they assumed a basic half-wave dipole antenna with no handset and 15 mm from the head surface. The results showed that the model's form and size are unimportant, and that the volume-averaged spatial peak SAR produced with the homogeneous phantom is only marginally better than the worst-case exposure in the inhomogeneous ones.

The crucial point that this paper quotes from [61] is that the power absorption and output power do not have a proportionate connection. It is, however, proportional to the antenna current squared. Furthermore, the square of the H-field incident on the skull can be used to calculate the induced SAR. For antennas close to the head, the H-field decays about l/r near the antenna, and this dominates the rate of field decay in the first few centimetres of the head, while the typical attenuation by absorption in the head takes control beyond that, as shown in Figure 5.

Fujiwara et al. [62] attempted to improve the modelling of the head's epidermal layer by providing a new Debye model for its permittivity in early 1997. They reasoned that because of its relatively high permittivity in proportion to fat, its potential role in screening interior structures had been overlooked. The discrepancies in the estimated SAR (using plane-wave incidence at 1.5 GHz) were, on the other hand, on the order of a few percent.

This effort looks to be more detailed than what is required to improve the designs of working handsets.

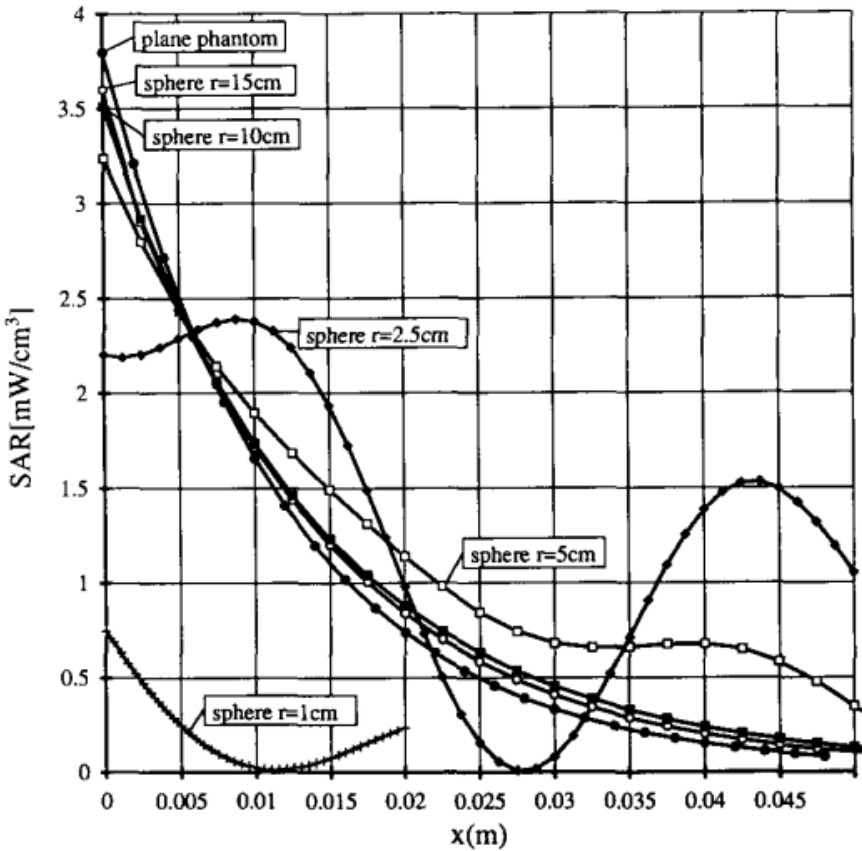

**Figure 5.** The absorption of the plane phantom is compared with the absorption in spheres of different radii, T, consisting of and using the same λ/2 dipole excitation (840 MHz. d = 25 mm). All values are calibrated to an antenna current of 100 mA.

Meier et al. [63] duplicated the investigation of Hombach et al. [35], using 1.8 GHz to determine if the homogenous head models were adequate for this purpose. Except for the frequency adjustment, most features of the phantoms, antennas, and modelling remained the same. For the complicated head models, the finite integration technique (FIT) programme, MAFIA, was used, and for the spherical models, the multiple multipole method was used.

The impact of several layered topologies (for instance, bone–brain and skin–bone) on the spatially averaged SAR was investigated and compared to homogeneous models. Although complicated oscillations in the SAR were discovered inside the inhomogeneous surface layers, it was concluded that using a homogeneous head model ($\varepsilon_r' = 41$, $\sigma = 1.65$ S/m) for the frequencies 1.5 GHz–2.5 GHz, likely overstated the exposure. The overestimation for the 10 g averaged SAR was <5%, whereas the overestimation for the 1 g averaged SAR was <20%. As a result, worst-case scenarios can be satisfied utilising homogeneous head models.

These two publications provide sufficient justification for the use of simple homogeneous, spherical models throughout the current study when the major goal is antenna design optimisation, rather than the high precision in SAR evaluation.

Anatomical structure data files were developed and integrated into field solution software, but, until recently, the models of handsets employed in the computations were crude approximations of commercial sets, often consisting of a plastic-covered rectangular metal box. Tinniswood et al. [64] published a paper in 1998 that examined the strategies for transferring CAD (computer-aided design) files of real handset designs into the

field solution with increased accuracy and less setup time. The primary strategy was to coat the interior profile of the box, which was defined as metal in the CAD file, in plastic. They also discussed how to make a parallel FDTD technique that allowed for distributed processing and improved resolution. The research provides useful information on the required resolution for modelling by comparing findings from a model with $1.974 \times 1.974 \times 3.0$ mm voxels, to 2 additional models with voxels that are around 1 mm cubes. The 1 g averaged SARs and total absorption varied slightly across the three models. The peak 1 g SAR (for a "box" telephone) is estimated to be roughly 12.64 kg$^{-1}$, while the peak 1-voxel value is 3.7 and 6.8 times more than the lower resolution model, and 1 of the high-resolution models, respectively. This demonstrates that "point" SARs significantly outperform values averaged over a 1 g cell. This is explained in part by the fact that the attenuation of a plane wave in brain tissue is roughly 3 dB/cm, as shown above.

According to other researchers, the highest 1 g SAR value is assumed to occur in the ear, and the peak 1 g value for the brain is only about 2.2 kg$^{-1}$. The CAD file telephone's peak and average SAR values are a few percent lower on average, according to the study, which can be explained by its contoured design that allows for less contact with the head. Small elements, such as the plastic operational buttons on the TV, are demonstrated to have little impact. SARs were only increased by 6% when the metallisation inside the "actual" telephone was reduced from 13.5 cm to 4.5 cm, whereas the plastic box was left intact, while the placement of a half-wave resonator within the box, just below the metal, had only a negligible effect. It was determined that the internal circuit board details would be insignificant and that there was little chance of unexpectedly resonant circuit components having a substantial impact.

In 1988, Kim and Rahmat-Samii [65] studied the SAR, antenna impedance, and patterns in the near- and far-field for a half-wave antenna at 900 MHz in the presence of human heads at a distance of 2 cm. The results of the FDTD approach in the presence of a one-layer homogeneous spherical head, three-layered spherical head, and anatomical head constituted of seven biological tissues were compared in this research utilising the MRI. A comparison of the obtained gain patterns of the half-wave antenna, in the case of homogeneous, anatomical, and three-layered spherical heads, has been carried out in this work. It is noted that an agreement was observed from the forward gain patterns from the three above-mentioned configurations, while the anatomical head is approximately 3 dB lower than that with the three-layered head in the backward gain scenario. The slightly lower gain that is presented with the anatomical head can exist due to the non-spherical head shape. Conversely, the spherical head model produces very similar radiation patterns as well as foresees the correct null locations.

The three-layered and homogeneous head field distributions are comparable. The strength of felid in the anatomical head was found to be lower compared to others. The important observation states that all the strengths of the electrical field close to the head surface in all three heads scenarios are in reasonable agreement. Practically, achieving a precise estimation of peak SAR can be attained by exploiting even a simple homogeneous spherical head model for a different number of antenna structures.

The combination of FDTD with the MoM (metal–oxide–metal) method was investigated by [66]. This combination is utilised for computing the electromagnetic fields of shielded RF coils that are loaded with a structurally precise human head model for use in high-frequency MRI applications. The magnetic field and SAR (known as the B1 field) that are excited through any RF coils can be predicted. The outcomes of the B1 field distribution and SAR that are excited through shielded and end-capped birdcage coils were computed at several frequencies, namely 64 MHz, 128 MHz, 171 MHz, and 256 MHz. The obtained outcomes illustrate that when the frequency of the B1 field increases, the SAR value increases as well, and a strong inhomogeneity at high frequencies may be exhibited by the B1 field.

Fujiwara et al. [67] used a thermal model of the head that included blood flow and conduction to try to convert a computed SAR into a temperature rise. Both the electromagnetic

and heat flow problems were solved using the FDTD approach. A plane wave at 1.5 GHz, incident broadside to the face, with a power density of $S_m$ (W/cm$^2$) was the incident field, as represented in Figure 6. The largest steady-state temperature rises in ocular and muscle tissue were around 0.35 °C, with the majority of the rise occurring within 2 min of exposure. The peak brain temperature rose at a consistent rate of 0.18 °C after starting at 0.024 °C. The final temperature rise in the hypothalamus was only 0.0029 °C, which was two orders of magnitude lower than the temperature regulating response threshold. These authors seemed to believe that only thermal impacts may be biologically important. This work is planned to be extended to near-field sources in a short amount of time. In the same year, Mangoud and his colleagues [68] presented a type of phased array antenna for cell phones based on a hybrid MoM/FDTD method. The idea behind such a design was to produce spatial null in the near-field zone and in the same direction as the human head. For the overall performance, in terms of efficiencies, the azimuth coverage was enhanced, and the peak SAR within the head was also improved by around 10 dB. Later, the work was improved to increase the isolation between the antenna and human head model [69].

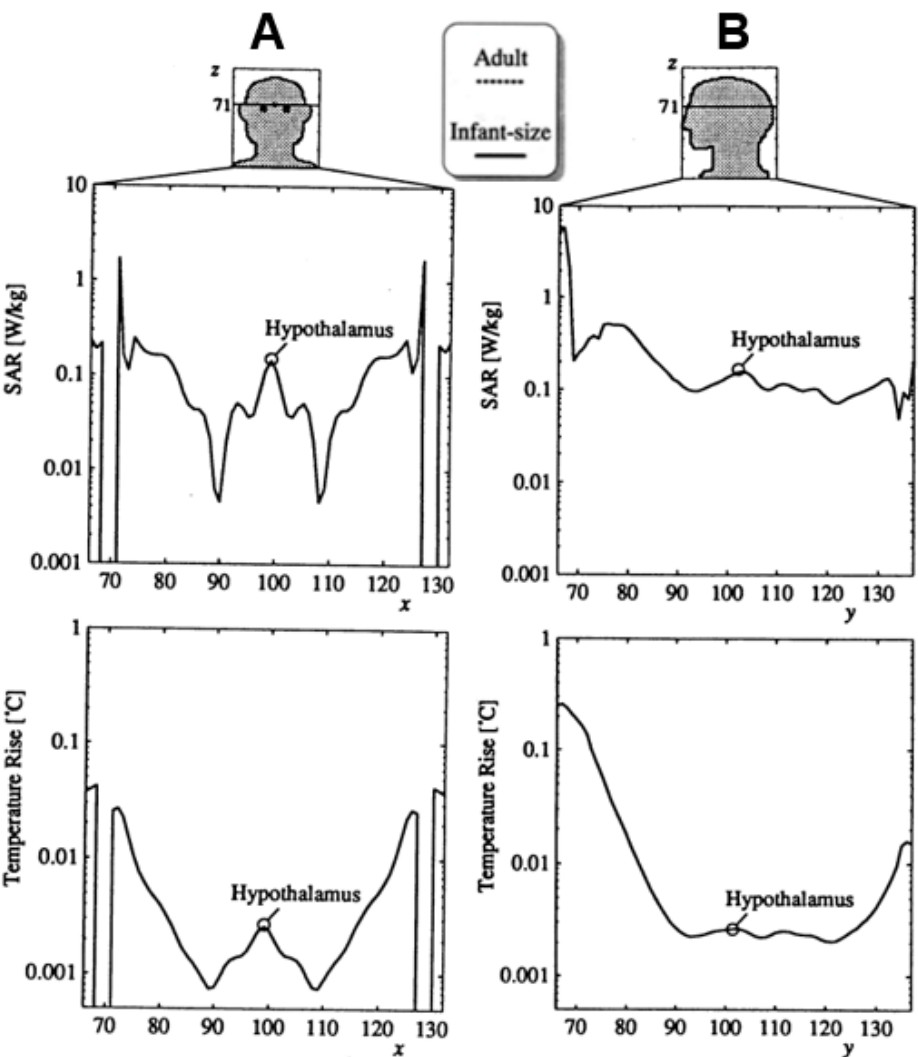

**Figure 6.** Partial distribution of (**A**) SAR and (**B**) temperature rise in the line passing through the centre of realistic head models, both from front to the back side and from the right to the left side [65].

In 2000, Iskander [70] employed the multigrid FDTD code (which is more efficient and memory-efficient than FDTD) to calculate the SAR values and distributions in the human skull, while taking the influence of the human body into account. During this calculation,

two antenna layouts were applied: monopoles along with a metal box, and a PIFA type; the used frequencies were 900 MHz and 1900 MHz, respectively. Additionally, other calculations were being conducted to investigate the impact of the antenna positioning on the calculated SAR values. In the case of a vertically positioned monopole antenna at 900 MHz, and from the accomplished outcomes, it was demonstrated that overlooking the impact of the human body can cause an underestimation of SAR values by around 53% when a 4 cm distance was obtained between the handheld phone and hand. In the situation of using a human head that is isolated from the neck, accurate SAR results can be attained, while the distance of 0.5 cm was obtained between the head and handheld phone. Reasonable calculations for both the scenarios with and without humans were stated at the 1900 MHz frequency of interest. In the second type of antenna (PIFA), the SAR results were less than the ones in the first antenna type (monopole). A small variation was obtained in the vertical orientation while the human body was present. The obtained results in the case of the horizontally orientated antenna were smaller when human body effects were taken into consideration.

In 2001, V. Perez et al. [71] divided their study into three simulations. These three simulations produced completely different goals. The primer simulation was carried out for the purpose of analysis, for which its parameters were deemed to be the most influential in the case of measuring the SAR. Such a simulation process verified that the exploitation of additional mathematical models does not lead to such excessive change in the values of the SAR, while, on the contrary, including the model of the hand demonstrated a reduction in the SAR, but was still not considered as significant (roughly 10%). For the utilisation of both bands, namely 900 MHz or 1800 MHz, it was decided that certain factors, such as the higher frequencies, higher vicinity to the human head, as well as the compact volume of the terminal, could lead to unacceptable SAR outcomes. The second type of simulation was built up to at least minimise the values of SAR that are accomplished in the worst scenarios. This can be achieved by exploiting many absorbent materials. Dielectric elements with several configurations were employed, such as an angular section of a cylinder, half a cone, and half a cylinder. These layouts may be positioned near to or around 2 mm away from the monopole antenna, which will contribute to modifying their thickness. The third simulation type compares the performance of a specific microstrip antenna, described in [72], to that of a standard monopole antenna. The microstrip antennas produced better results. The mobile terminal was designed as a solid metal box with a/4 monopole positioned on top of it, acting as a radiating element in this project. A series current generator was used to feed the monopole antenna, which produced a continuous sinusoidal signal with a peak value of 0.1 A.

Mangoud et al. [68] used the MoM–FDTD hybrid technique to investigate the effect of humans on the polarisation purity of several types of circularly polarised handset antennas for personal satellite communications. SAR in the head assessments was completed. Promising outcomes were attained by applying the above approach, according to the physical expectations.

In the case of designing a monopole antenna on a box case along with several materials, studying the maximum SAR that is said to exist on the head was performed by H. Chen and W. Chiou [73] in 2002. The obtained results within the above investigation are in comparison with the ones that were achieved from the mobile dipole antenna that considered doubling the length of the monopole structure. This research was conducted utilising the FDTD method at 900 MHz, with 2.5 cm between the head and the antenna. Higher SAR values were achieved when materials with larger dielectric constants and conductivities were used in the cellular phone's box casing.

Later, in 2002, Jung and Lee [74] investigated the impact of placing a ferrite sheet on the handset. They proposed SAR estimation and reduction approaches were both based on power conservation. It was demonstrated that attaching ferrite material to the handset may contribute to reducing not only the SAR, but also such a material can minimise the radiated power. It is also said that the SAR can be greatly and considerably mitigated, more than the

radiated power. Therefore, the ferrite may be deemed as one of the convenient approaches to reduce the SAR.

The chassis influences on the SAR characteristics of the handset antenna were investigated numerically with FDTD software by Dou [75] in 2002. A reduction of about 50% of the absorbed energy in the head was observed; this was reached by the optimisations of the chassis size, while the used antenna type was not modified or changed. The ideal length of around 126~180 mm was used for the 900 MHz and a length of 48~84 mm in the case of 2 GHz. At both the ideal above-mentioned lengths, the 1 g SAR values at 900 MHz were around 0.87~1.5 W/kg for 600 mW input power of the antenna. In the second scenario and for the same power, the SARs were practically doubled up at the targeted frequency of 2 GHz. Furthermore, the obtained outcomes demonstrate that the area that is on the right side of the ear for both given spectrums is where most of the absorbed energy is concentrated, independently of the chassis dimension, and the efficiency is much better at the higher frequency.

Al-Nuaimi et al. [76] used an anatomically based numerical head/shoulders model to perform FDTD simulations. For both a monopole and a PIFA fed by a CW input power of 1 W, the evaluations of mass-averaged SAR and efficiency were reported, as well as the effect of the handset height. The peak SAR was reduced by 30% at handset heights of approximately $\lambda/2$ and $\lambda/4$, according to research. The peak SAR was reduced by approximately 1 W/kg, by rotating the handset into the normally speaking position (60°).

The spatial distribution of SAR inside a human head model was computed by Kuo and Kuo [77] in 2003, as the ohmic loss averaged over 1 g or 10 g of tissues. Then, using an explicit finite-difference formulation of the bioheat equation, the thermal response as a function of time was recorded until the steady-state temperature distribution inside the skull was reached. This variant used a handheld phone with a quarter-wavelength monopole antenna that worked at 900 MHz and 1.8 GHz. The antenna output powers were tuned at 0.6 W at 900 MHz and 0.125 W at 1.8 GHz, respectively. The temperature distribution, T = T(x, y, z: t), within the head model was achieved through the bioheat equation:

$$K\nabla^2 T + \rho.SAR - B(T - T_b) = C\rho\frac{\partial T}{\partial t} \tag{6}$$

where K is the thermal conductivity [J/(sm °C)], SAR is the input EM heating source, *B* is in proportional relation to the blood perfusion [J/(sm$^3$.°C)], $T_b$ is the blood temperature and finally, C [J/(kg °C)] and $\rho$ [kg/m$^3$] is the tissue-specific heat and density, respectively. The brain's steady-state temperature rise was calculated to be 0.1° at 900 MHz and 0.03° at 1.8 GHz.

In 2004, Kivekas et al. [78] presented a crucial problem with the antenna for internal use in mobile phones. Within this study, the impact of chassis parameters, such as width, length, distance, thickness from the phone to the head on the efficiencies, bandwidth, as well as the SAR values of the proposed patch design, were examined. The combination of the antenna–chassis was in the actual phone-use position alongside an anatomical head model. Moreover, their study produced two different hand models.

A 3D sub-gridding technique was investigated in 2005 [79]. An approach of FDTD to model such isolated cochlea for SAR distribution, which is exposed to handset radiation at the spectrum of 1700 MHz, was proposed [79]. An increasing spatial resolution of 1, 1/3, 1/5, and 1/7 mm was employed to model the cochlea. The initial simulation process was executed at the spatial resolution of 1 mm. An increment of 1/3, 1/5, and 1/7 of the numerical resolution utilising sub-gridding was met without altering the spatial resolution. The next processes of simulations were conducted, while the joint increase in both numerical and spatial resolutions was achieved. Looking at the accomplished results, it can be concluded that sub-gridding is vital only in the case of increasing both the numerical resolution of the computational grid and the spatial resolution of the model.

In 2008, a method of FDTD was used to determine the SAR distribution in a human head near a handheld cellular phone [80]. The biological effects of radio frequency radiation are proportional to the rate at which it is absorbed. In this paper, a discrete human head

model was combined with a discrete handheld receiver model in an FDTD volume. For a digital european cordless telephone (DECT) system operating at 1.8 GHz, FDTD simulations were run. Over the human skull, SAR distributions were determined. Other examples of FDTD implementation for SAR predictions can be found in [37,81–85].

In 2009, Xu with his research team [86], proposed the study of SAR and near fields of ingestible wireless devices (IWDs) in the case of two models of the human body, in which the values of their dielectrics were improved to ±10%, compared to the original, and ±20% exploited the approach of using FDTD. A comparison of the radiation characteristics of the IWDs in the scenario of the human body models was made at 800 MHz, when the dielectric values were changed and kept as original, as illustrated in Figure 7. A variation of radiation intensity outcome of around 1.6 dB, which is closer to the abdomen surface within the 20% variation of dielectric values, was demonstrated. Increasing the conductivities of human body tissues proportionally leads to an increase in SAR, while occasionally the increase in the relative permittivity of human body tissues results in a decrease in SAR values. Following IEEE safety standards, the IWD was said to be secure to be exploited at an input power of less than 9.3 Mw, to the degree that the compliance of safety was concerned; later, the technique was used at a 430 MHz operating frequency [87], at 1200 MHz [88], and a 2 GHz mobile phone frequency band [89].

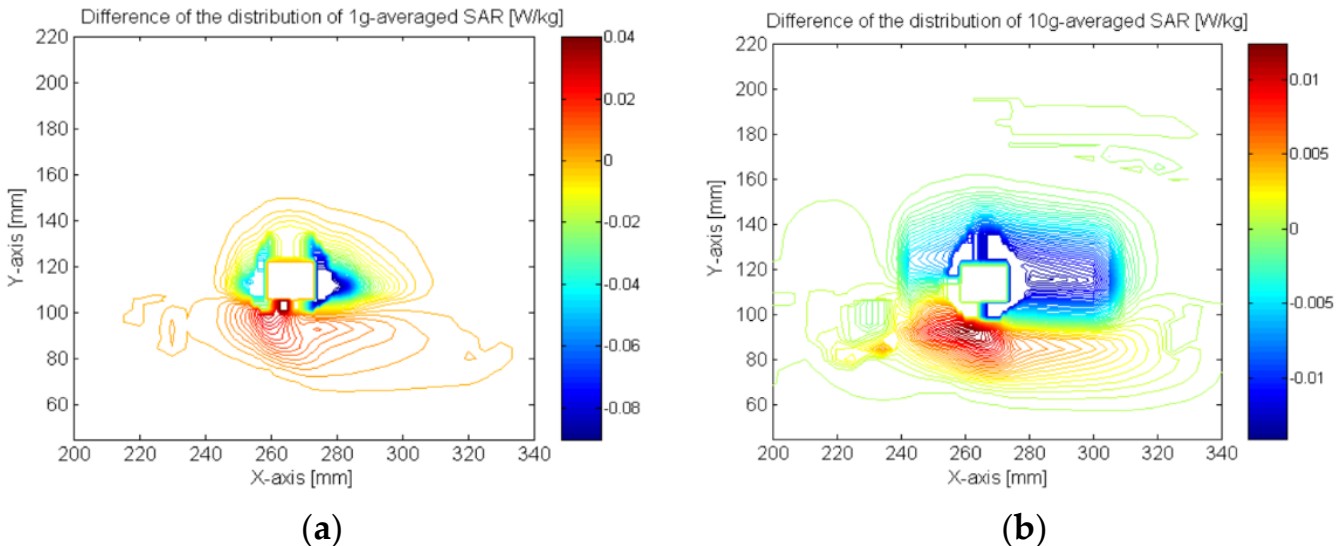

(**a**)    (**b**)

**Figure 7.** (**a**) Difference of the distribution of 1 g averaged SAR between two levels of conductivity: 20% increase and original at the frequency of 800 MHz with X orientation; (**b**) difference of the distribution of 10 g averaged SAR between two levels of conductivity: 20% increase and original at the frequency of 800 MHz with X orientation [84].

In 2011, Lee and Yun [90] discovered that for smartphone sensitivity at 835 and 1900 MHz, the SARs in 5- and 9-year-old European and 7-year-old Korean head models for children were studied in comparison to the SAM (specified anthropomorphic mannequin) phantom. During the examination, the 5- and 7-year-old child models' pinnae were also compressed. The cheek and tilt locations that happen when such a phone's earpiece is located at the EEC (ear entrance canal) were compared to the same areas that appeared when the earpiece was positioned at the ERP (Ear Reference Point). SAR changes in the tissue region surrounding the phone were studied for varied fat and skin characteristics, as well as distinct interior muscle and fat morphologies. For every frequency, the SAR was calculated using a simple monopole antenna phone model. At 1900 MHz, the SAR values in the child models were higher than the SAM phantom, thus a phone model with a planar reversed F antenna was also used for validation. Every tissue, including the pinnae, had its 10 g SAR values harmonised to a forward power of 1 W at the phone's feed point. There are four points for the outcomes, which are:

- At 835 MHz, a compressed pinna did not exhibit significant changes in the SAR values, but, at 1900 MHz, there was also an average 25% improvement in regional peak 10 g SARs for pinna-excluded tissue and a 29% rise for pinna-included tissue.
- As compared to the ERP location, a phone earpiece positioned at the EEC generated greater SARs in several situations.
- When the smartphone was touched, the peak 10 g SAR was shown to be extremely sensitive to the muscle and fat structure under the skin; a tendon interior head framework resulted in a greater peak of 10 g SAR.
- The SAM phantom does not appear to offer a reasonable figure of child head radiation at 1900 MHz; 45 percent (pinna-excluded: IEEE Std C95.1) and 75 percent (pinna-included: ICNIRP recommendations) of the 40-focus area they evaluated exhibited greater SAR findings than the SAM phantom.

In contrast, Gosselin et al. [91] determined that the formulae may be utilised for adults standing inside the emitting field region of base-station antennas working at frequencies ranging from 300 MHz to 5 GHz and at distances greater than 20 cm, as demonstrated in Figure 8. It was demonstrated that the absorption process and scientific data of the height, weight, and body weight of the population may well match the 95th percentile absorption for the population. Three different anatomical human models (Thelonious, Duke, and Ella) were statistically validated by exposing them to 12 general base-station antennas in the wavelength region from 300 MHz to 5 GHz, at 6 different distances ranging from 1 cm to 300 cm. The proposed methods for adult models are cautious in estimating the SAR exposure values of the two adults, but not of the child, out of the 432 analysed combinations.

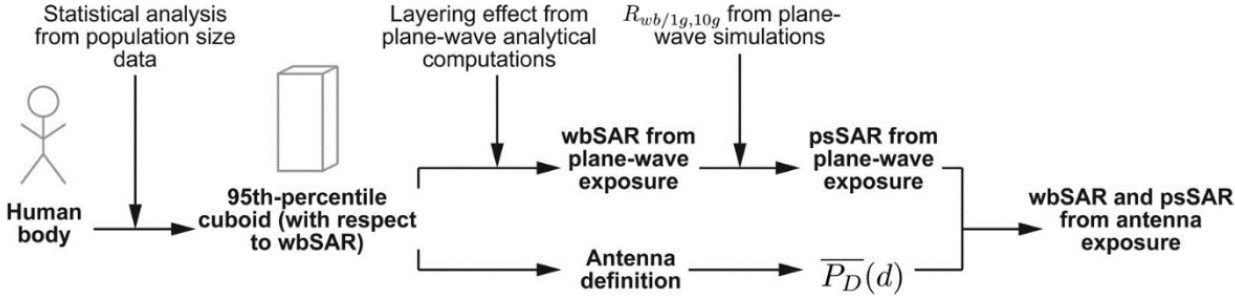

**Figure 8.** Diagram representing the steps in elaborating the SAR estimation formulas.

In 2012, Okano and Shimoji [92] advanced the measurements compared between a thermal SAR measuring system and an electric field probe scanning system. The findings of the measurement procedures were clarified by utilising a regular parameter setting; arbitrary measurement settings were not modified. The Dosimetric Assessment System Ver. 4 (DASY-4) was used to scan the electric field probes. As the thermal SAR measurement system, the authors' TESS (thermal evaluation system for SAR) was employed. A liquid phantom in line with International Electrotechnical Commission (IEC-62209) [93] was utilised to evaluate the DASY-4 measurement. DASY-4 measures SAR at 2–3 and 5–6 GHz frequency bands. In comparison, for the microwave exposure detected by TESS, a new semisolid phantom corresponding to the living tissues was created (also by IEC-62209). SAR is constantly recorded from 2 to 6 GHz at about 0.5 a GHz periods since TESS sensors do not have frequency dependence. Other examples of experimental assessments using room electromagnetics and simultaneous electric field and temperature measurement predictions were published in [94,95].

Later, over a frequency range of 900 MHz to 3.7 GHz, Li et al. [96] investigated the absorption of electromagnetic fields in the hand. The envelope of the peak spatial absorption rate in the hand may therefore be determined. It also serves as a foundation for developing measurement techniques for determining if smart networks in the hands follow certain absorption rate restrictions. Plane waves and dipole antennas were both utilised to

examine the patterns of RF absorption in finger tissue and hand models for both near-field and far-field exposures. The outcomes show that there are absorption improvements in the hand, which are not found in a conventional flat phantom. Depending on the model settings, improvements of several dB are shown. Based on the flat phantom measurements, a method for conservatively estimating exposure in the hand is presented. In addition, Zhou et al. [97–100] adopted a rectangular head phantom and DASY-4 near-field probe, as depicted in Figure 9, to measure the SAR performance of the proposed balanced folded dipole antenna. There is an excellent agreement between the simulated and measured SAR of the antenna at 1.8 GHz mobile operating frequency.

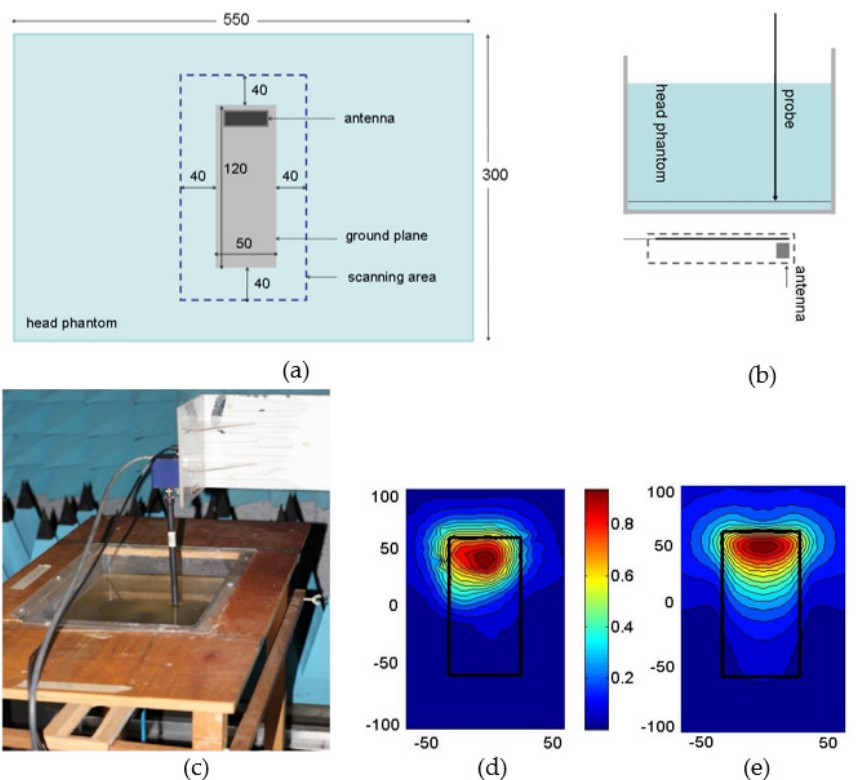

**Figure 9.** Experimental setup and SAR measurement, (**a**) bottom view; (**b**) side view; (**c**) practical photo; (**d**) measured normalised SAR; and (**e**) computed normalised SAR at 1.8 GHz.

Thotahewa et al. [101] reported the electromagnetic impacts of head-implantable transmitting electronic devices using UWB (ultra wideband) wireless technology based on an impulse radio. When implanted in a human head model, the simulations showed the performance of an implantable UWB antenna set to function at 4 GHz with a 10-dB bandwidth of roughly 1 GHz. To examine the transmitting device's compliance with international safety requirements, the SAR, SA (specific absorption), and temperature rise were studied. The age dependence and frequency of tissue characteristics, including the absorption coefficient, were considered. The human head's SA/SAR fluctuation is shown with varying signal bandwidths, input power, and antenna orientations. The SA and SAR values for the highest peak power limitations of 27.4 and 41.3 dBm/MHz, respectively, are substantially within the ICNIRP permitted values. The temperature rise caused by exposing the head tissues to the IR-UWB (infrared ultra wideband) electromagnetic field at such peak power limitations was determined to be well within the range of the human body's thermal regulating mechanisms. The SAR was also shown to be strongly dependent on the bandwidth of the IR-UWB signal. In addition, it is also possible to stimulate the antenna with a signal power greater than the outside permissible power limit of −41.3 dBm/MHz. It was discovered that a pulse with a peak power limit that is 13.9 dB greater than the FCC

authorised peak power, may be used without breaking SA/SAR limitations or the outside FCC standards for this specific model.

In 2014, Monebhurrun et al. [102] investigated an interlaboratory comparison of SAR computations utilising a CAD (computer-aided design)-based model of a commercial dual-band smartphone carried out by seven worldwide laboratories at 890 MHz and 1.75 GHz. The peak 10 g average SAR and return loss data collected with six various electromagnetic solvers were evaluated, despite variations in the approaches utilised in the solvers (whether the frequency or time domain) utilised for the interlaboratory evaluation. The results determine that the second frequency range of the dual-band smartphone exhibits considerably greater deviations. This might be because the collaborating laboratories used various mesh densities. Assuming the uncertainty budgets of both techniques, the numerical simulation findings are likewise in good agreement with the observations.

Furthermore, Ferikoğlu et al. [103] presented PSPICE (personal simulation program with integrated circuit emphasis), which, for the first time, was utilised for associated temperature increases, and SAR, using the emulation of an electromagnetic radiological, as well as the computation of SAR and temperature changes for a multilayer living tissue prototype at 915 MHz and 2.45 GHz. The findings are similar to fundamental field theory and demonstrate the PSPICE simulator's calculative and illustrative capabilities. This simulation technique is well suited to modelling layered human tissues in a visible and flexible manner, allowing for the easy observation of electric field changes at specified places of the open tissues, which may be created at desired periods.

In [104–106], respectively, were published other examples of a variety of NPs (nanoparticles) that have ferromagnetic characteristics at room temperature distributed in water; several two-stage techniques that include various nonlinear programming and initialisations methods, as well as including the stringent SAR and hardware restrictions directly; and a hybrid local SAR simulation approach for high-field MRI coil arrays that showed real-time simulation speeds and good numerical accuracy.

In 2015, Lee et al. [107] evaluated the SAM's conservatism by quantitatively comparing the SAR in the SAM with those in four anatomical head models at various ages for exposure from a common bar-type smartphone. At 835 MHz and 1.85 GHz, a numerical bar-type smartphone model with an inbuilt antenna at the bottom of the mobile body was implemented. This device has an SAR pattern and values that are comparable to the commercial bar smartphones produced in Korea. Spatial peak 1 and 10 g SARs were computed for both the SAM phantom and anatomical head models for two standard test locations. When the antenna is placed on top of the phone, the SARs are also computed. According to the findings, the SAM phantom presents a reasonable estimation for a phone model with the antenna on top. As the antenna is placed at the bottom, the hotspot in the SAM occurs further away from the antenna feed point, resulting in poorer 1 and 10 g SAR values when contrasted to the physical models. Another example of the particle swarm optimisation technique is used in [108] to determine the best position of the relay node.

In 2016, the modelling of the human head was performed in [109] using ANSYS HFSS (high-frequency structure simulator). For the simulation, the contemporary smartphone design was utilised. Due to the exposure to radiation from a smartphone working in the GSM (Global System for Mobile Communications)-1800 frequency range, the SAR distribution was evaluated for various layers of the skull. Due to their low conductivity, shields made of conductors decreased the performance of smartphone antennas, but shields built using insulators did not absorb any radiation. Components with conductivities ranging from 0.1 to 3 S/m are effective as smartphone shields. Components with comparable characteristics that occur in the actual world are sought for depending on these attributes. Germanium and skin (leather) are two examples of substances in this area. Additionally, it was discovered that the shields may be placed either outside or inside of the smartphone, resulting in the same efficiency of the smartphone antenna and SAR reduction. A possibility for the real use might be to use the shield as a smartphone flip cover case made of germanium or leather.

Based on the fixed location SAR/PLD (power loss density) model derivation and the Fourier series expansion, Li et al. [110] calculated a high-order SAR/PLD model for estimating the electromagnetic absorption caused by multiple transmitters in portable devices. This model is used to evaluate, estimate, and compute the maximum EA generated by multiantenna transmitters, as shown in Figure 10. To verify the efficiency and resilience of the proposed SAR/PLD model, four computational and one experimental sample are provided. Both the simulated and measured results show that when the model order is set to three, the high-order model achieves the highest accuracy in calculating and predicting the maximum SAR/PLD values. It has also been shown that the high-order model functions efficiently in the LTE bands of 1800, 2300, and 2500 MHz, as well as in the 5G and millimetre–wave communication frequency bands of 2800 MHz and 100 GHz. It also operates well for varied numbers of transmitting gestures, antennas, and geometries, indicating the suggested model's remarkable resilience. The suggested SAR/PLD model, which is capable of properly calculating and forecasting SAR/PLD values, may be utilised to improve the bio-electromagnetic performances of multiantenna smart devices when paired with other telecommunications.

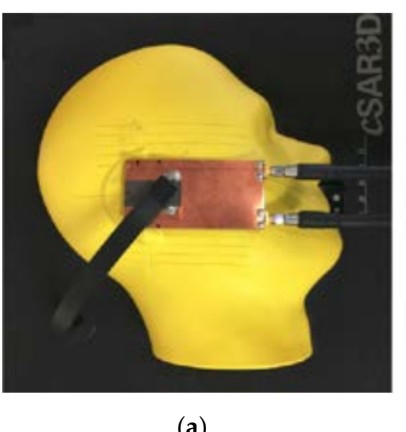
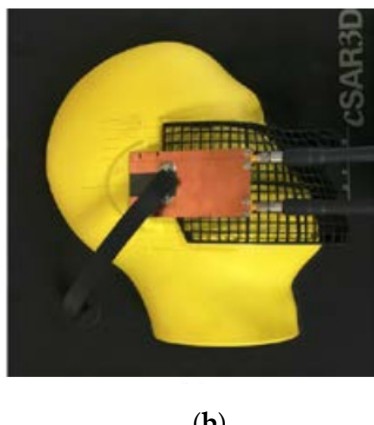

(**a**)            (**b**)

**Figure 10.** Two-port antenna board placed next to the c-SAR3D SAM phantom with (**a**) touch gesture and (**b**) 15 degree tilt gesture.

Later, in [111], Okano et al. offered a thermal SAR assessment approach that may be utilised in the 300 MHz to 6000 MHz bandwidths, which is commonly utilised by smartphones. For the first time, the peak of the SAR was validated using this continuous broadband test at 1800 MHz, as shown in Figure 11. Furthermore, the mean value of every evaluated datum indicated a deflection of about 10% to 20% throughout the whole frequency range, as a comparative measurement result with the electric field probe system and the suggested system. Other examples of a quick and extremely accurate estimate technique for determining the SAR values for arbitrary relative phase combinations and the SAR 10 g fluctuation in the human torso when a phone is held at varied locations or tilt degrees for data transmission, were published in [112,113], respectively.

In 2018, Taguchi et al. [114] compared the internal physical parameters of a human model exposed to an intermediate frequency electromagnetic field frequency ranging from 100 kHz–10 MHz. The fact that multiple computational assumptions were used, such as full-wave analysis and quasistatic estimation, was a distinguishing aspect of this intercomparison. The maximum frequency at which the quasi-static assumption was viable was specified as being less than 30 MHz for free-standing humans and 10 MHz for grounded people. Then, from 100 kHz to 100 MHz, the authors calculated the connection between the external field strength and local SARs, as depicted in Figure 12. The SARs for the magnetic field exposure ranged from 100 kHz to 10 MHz, as well as the local SARs in the grounded limb. Then, they proposed that the ICNIRP's reference level was highly cautious for frequencies less than 30 MHz, while the IEEE C95.1 presented some marginal

disagreement. Another observation was that local SARs were typically satisfied, even for the RL built for the whole averaged SARs in both the electric and magnetic fields. The data acquired here might be beneficial in updating and reconciling the differences in the reference values established by IEEE and ICNIRP.

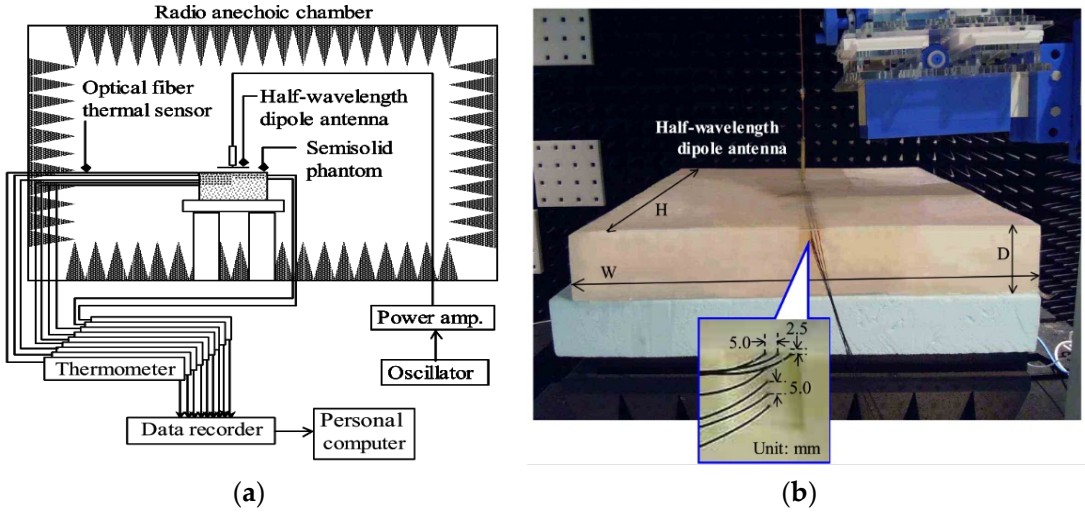

**Figure 11.** Thermal evaluation system for specific absorption rates: (**a**) system diagram, and (**b**) rectangular box-shaped semisolid phantom used for the measurements [109].

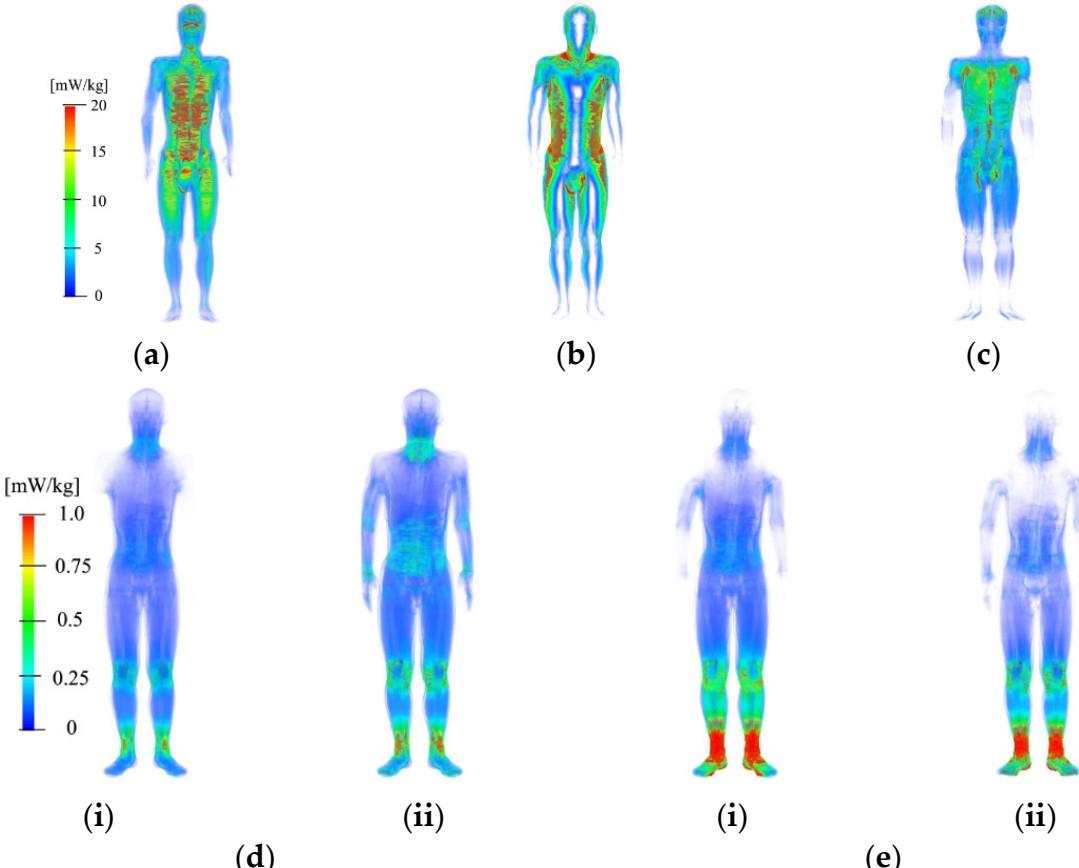

**Figure 12.** SAR distributions in TARO (Japanese adult male model) standing in free space for magnetic field exposure at 10 MHz. The exposure directions are (**a**) LAT, (**b**) AP, (**c**) TOP, (**d**) free space at 60 MHz, and (**e**) grounded at 30 MHz for (**i**) electric field, and (**ii**) plane wave exposure.

In 2019, Nagaoka et al. [115] presented a new child voxel model that was constructed and tuned to the ICRP (International Commission on Radiological Protection) reference values, such as 10-year-old children. The WBA-SARs (whole-body averaged specific absorption ratios) of these children's voxel models were calculated under whole-body exposure conditions, with H-polarisation and E-polarisation plane wave exposures ranging from 10 MHz to 6 GHz. When corrected to the ICRP's reference values, the WBA-SARs for the child voxel models were 40% to 60% higher than for the adult voxel models. The WBA-SARs of the child voxel models exceeded the fundamental restriction by no more than 20% (or 0.08 W/kg) for the entire environment, when the incident power density was adjusted to the reference level for every band of the ICNIRP's RF safety standards. Another example of a stochastic technique, known as GS-FDTD (geometrically stochastic finite-difference time-domain), to directly include the statistical fluctuations of model geometry into the FDTD approach was published in [116].

The [112] Kour et al. study aimed to shed light on the health effects of millimetre wave frequencies, including thermal heating in body tissues as temperatures rise, power density, and SAR. As a solution, a methods were proposed for electromagnetic radiation reductions in mobile communication systems, including greater energy efficiency, thermal radiation mode, advocating its safe use, encouraging green WCNs (wireless communication networks), and decreased complexity for future generations. In the form of simulation data, the concept also confirms the lower SAR, power density, and temperature rise produced in human tissue when compared to previous models. It has the potential to improve the safety and dependability of 5G and, in the future, 6G networks. In [117–119], respectively, were published other examples of a new modern multi-band PIFA antenna that covers four frequency spectrums, such as WIMAX (worldwide interoperability for microwave access), 4G LTE (long-term evolution) GSM and Wi-Fi/Bluetooth; a CPW-fed (co-planar waveguide), small, CRLH (composite right/left-handed)-based antenna supported by an all-textile $2 \times 2$ AMC (artificial magnetic conductor) array that was presented as an integrated design; and a comprehensive mathematical model of a human hand that was used to compute the dosimetry of an SAR and the temperature rise caused by touch contact currents in the frequency range of 100 kHz to 100 MHz.

In 2021, Hossain et al. [120] presented a planar $2 \times 2$ MIMO (multiple input and multiple output) arrangement for a 5G mobile that was proposed. To examine the feasibility of several sub-6 GHz frequency bands in future 5G communication, a single element MPTPS (modified planar tree profile shape) antenna was constructed. The single MPTPS antenna measured $40 \times 25$ mm$^2$ in size. This antenna's tuning was accomplished using the PGP (partial ground plane) and EBG (electronic band gap) methods. The antenna operated between 2.81 and 7.23 GHz and had a (VSWR (voltage standing wave ratio) < 2) bandwidth of 4.42 GHz, covering all mid-range sub-6 GHz 5G frequencies. It also boasted a decent gain of 3.14 dBi, high accuracy of 96 percent, and a bi-directional emission pattern. The antenna was built inside a $145 \times 75$ mm$^2$ mobile mainboard and used in a MIMO configuration with polarisation diversity, as shown in Figure 13a. There was more than $-21.1$ dB isolation between the separate ports. For the MIMO in the band, a good gain of up to 6.59 dBi was recorded. In terms of MIMO performance, a good envelope correlation coefficient of less than 0.0029 and a minimum diversity gain of 9.9853 was found. The research was expanded by including an LCD (liquid crystal display) for enhanced strength, and a hand phantom to evaluate the results in terms of SAR. At 3.5 GHz, the SAR value is reported to be as low as 0.887641, as shown in Figure 13b. This concept spurs the researcher to create high-performance MIMO arrays for 5G devices. More interesting work and examples on studying and measuring the SAR of several antenna types and structures are reported in [121].

To summarize all the numerical and experimental EM radiation assessments, Table 2 presents all the techniques from 1978 to 2021 in term of operating frequencies and SAR values.

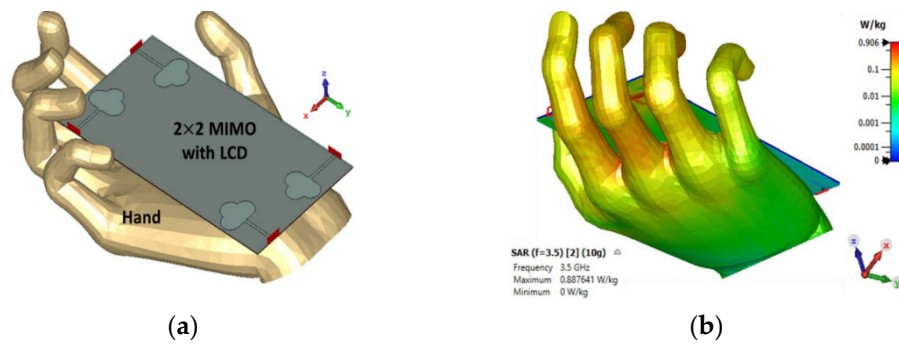

<div align="center">(<b>a</b>)          (<b>b</b>)</div>

**Figure 13.** The proposed 2 × 2 MIMO configuration for smartphones: (**a**) MIMO array with LCD and hand phantom, (**b**) AR values at 3.5 GHz [118].

**Table 2.** Summaries of all the techniques utilised between 1978 and 2021, including the frequency bandwidth and SAR value.

| References | Method | Frequency Range | SAR Value |
|:---:|:---:|:---:|:---:|
| [48] | Flat surface phantom, human head, and torso | 800–900 MHz | <0.3 mW/gr |
| [49] | Head and body phantoms | 50, 150, 450, and 800 MHz | 1–2 W/kg |
| [50] | Human body model and a tensor integral using the technique of moments | 350 MHz | SAR is varied by factors ranging from 3 to 10 |
| [51] | FDTD | 350 MHz | N/A |
| [54] | FDTD | 900 MHz and 1.9 GHz | 10 W/kg averaged |
| [55] | FDTD | 900 MHz and 1.8 GHz | At 900 MHz and 1.8 GHz, 10 g of tissue produces 2.1 and 3.0 W/kg, respectively. At 900 MHz and 1.8 GHz, 1 g of tissue produces 2.3 and 4.8 W/kg per W, respectively. |
| [37] | MMP | 835 and 1900 MHz | Several SAR values were covered in different areas of the body |
| [57,58] | FDTD | 915 MHz | For 1 W of supplied power, the head SAR peaks between 2.0 and 3.8 mW/g and averages between 0.08 and 0.10 mW/g |
| [59] | FDTD | 900 MHz and 1.5 GHz | N/A |
| [40] | FDTD | 900 MHz | When compared to the maximum value obtained by the inhomogeneous phantoms, the averaged spatial peak SAR values are tiny |
| [61] | 3D-MMP | Above 300 MHz | 8 mW/g in a controlled environment 1.6 mW/g in an uncontrolled environment |
| [62] | FDTD | 1.5 GHz | 5.5 W/Kg for the skin 0.395 for the whole head <5 W/Kg for the eyeball At 1 g average peak |
| [63] | FDTD | 1.5 and 2.5 GHz | <5% for the 10 g averaged spatial-peak SAR, and considerably <20% for the 1 g averaged value |
| [64] | FDTD | 1900 MHz | The peak 1 g SAR is estimated to be around 12.64 $kg^{-1}$, whereas the peak 1-voxel value is 3.7 times and 6.8 times higher than in the lower resolution model The brain's highest 1 g value is about 2.2 $kg^{-1}$ |
| [65] | FDTD | 900 MHz | N/A |
| [66] | FDTD with MoM | 64, 128, 171, and 256 MHz | The SAR value increases when the frequency of the B1 (magnetic field) field increases |
| [67] | FDTD | 1.5 GHz | Averaged SAR over any 1 g of tissues, such as bone = 1.33 W/Kg, brain = 2.27 W/Kg, muscle = 5.62 W/Kg, eyeball = 2.62 W/Kg, fat = 1.15 W/Kg, and skin = 4.47 W/Kg |

**Table 2.** *Cont.*

| References | Method | Frequency Range | SAR Value |
|---|---|---|---|
| [69] | Hybrid MoM/FDTD | 1.8–2 GH | One-element handset = 2.231 W/Kg, one-element handset with reflector sheet = 0.3814 W/Kg, two parallel dipoles = 1.2 W/Kg, two-element handset = 0.5166 W/Kg, and two-element handset with reflector sheet = 0.361 W/Kg |
| [70] | Multigrid FDTD | 900 MHz and 1.9 GHz | N/A |
| [71,72] | FDTD | 900 MHz or 1800 MHz | The 1 g spatial peak SAR value in the head was lowered by 70% when compared to the monopole |
| [73] | FDTD | 900 MHz | N/A |
| [74,75] | FDTD | 900 MHz and 2 GHz | At 900 MHz, the 1 g SAR values were determined to be between 0.87 and 1.5 W/kg At 2 GHz, the SARs are nearly doubled |
| [77] | FDTD | 900 MHz and 1.8 GHz | The 1 g average spatial peak SAR to 1.6 W/kg in the brain at 900 MHz The 10 g average spatial peak SAR to 2 W/Kg in the brain at 1.8 GHz |
| [78] | FDTD and MoM | 900 MHz and 1800 MHz | N/A |
| [79] | FDTD | 1750 MHz | N/A |
| [80–85] | FDTD | 1.8 GHz | N/A |
| [86–89] | FDTD and MoM | 430 MHz, 800 MHz, 1200 MHz, and 2 GHz | The 1 g and 10 g averaged SARs can reach 3.16 and 0.89 W/kg, respectively |
| [90] | FDTD | 835 and 1900 MHz | The spatial peak 10-g SAR of a 7-year-old Korean child head model (KR-7y) is 2.46 W/Kg, and the head average SAR is 0.203 W/Kg |
| [91] | FDTD and MoM | 300 MHz to 5 GHz | N/A |
| [92–95] | EFPS (electric field probe scanning) and FDTD | 2–3 and 5–6 GHz | N/A |
| [96] | FDTD | 900 MHz to 3.7 GHz | N/A |
| [101] | FDTD | 4 GHz | The SAR and SA results for the 41.3 and 27.4 dBm/MHz maximum peak power limits |
| [102] | EMPIRE based on the FDTD, EMPro based on the FDTD, HFSS based on the finite-element, microwave studio is based on the finite integration technique (FIT), Microstripes is based on CST's transmission line matrix approach, and Remcom's XFDTD is based on the transmission line matrix method | 890 MHz and 1.75 GHz | The peak 10 g average SAR at 890 MHz at right/check is 0.88 W/Kg from lab A, 0.90 W/Kg from lab B, and 1.06 W/Kg from the manufacturer. While the right/tilt is 0.44 W/Kg from lab A, 0.43 W/Kg from lab B, and 0.57 W/Kg from the manufacturer. The peak 10 g average SAR at 1.75 GHz at right/check is 0.80 W/Kg from lab A, 0.78 W/Kg from lab B, and 0.90 W/Kg from the manufacturer. While the right/tilt is 0.22 W/Kg from lab A, 0.23 W/Kg from lab B, and 0.28 W/Kg from the manufacturer |
| [103] | Lossy transmission-line device model existing in the PSPICE parts library | 915 MHz and 2.45 GHz | N/A |
| [107] | N/A | 835 MHz and 1.85 GHz | Both the SAM phantom and anatomical head models had their spatial peak 1 and 10 g SARs determined |
| [109] | ANSYS HFSS | 1.8 GHz | Maximum local SAR for the skin at a 0.1 cm thick shield is 4.85 W/Kg, while for the brain it is 0.95 W/Kg. Moreover, the maximum local SAR for the skin at a 0.2 cm thick shield is 3.9 W/Kg, while for the brain it is 0.75 W/Kg. |

**Table 2.** *Cont.*

| References | Method | Frequency Range | SAR Value |
|---|---|---|---|
| [110] | FDTD | 1.8, 2.3, 2.5, 28, and 100 GHz | Different SAR values |
| [111] | EFPS and FDTD ikaika | 300 MHz to 6000 MHz | The peak curve of the SAR 10 g has a maximum value of around 1.8 GHz |
| [114] | FEM (finite element method) | 100 kHz–10 MHz | N/A |
| [115] | FDTD | 10 MHz–6 GHz | The child voxel models' WBA (Whole Body Averaged)-SARs exceed the fundamental restriction by no more than 20% or 0.08 W/kg. |
| [120] | Finite integral technique (FIT) | 2.81–7.23 GHz | At 3.5 GHz, it has a value of 0.88 W/Kg, which is significantly lower than the acceptable standard value of 2 W/Kg, indicating that it is safe for use in proximity-centric applications involving human bodies |

## 6. Conclusions and Future Works

The radio frequency hazards, including the various safety standards and organisations due to the SAR, were presented in the present study. The modelling and measurements for the SAR values have also been reviewed and investigated during many years of work. The presented analysis of SAR increases the understanding of the joint behaviour of wireless and mobile devices with the human tissues, based on the induced field intensity and the dielectric properties of the biological tissues. The presented results show that the general trends of operational frequency, bandwidth based on SAR, could lead to an adequate approach to meet the actual performance of the new wireless and mobile devices.

The future works of this research direction of SAR can be summarised in the following areas: (i) simultaneous exposures to near- and far-field sources at different frequencies to define a new SAR measurement guideline and standard; (ii) studying MIMO antenna effects on SAR at millimetre-wave frequencies; (iii) quantifying the adverse (chronic) health effects from long-term RF exposure at 5G/6G mobile frequencies; (iv) finding the potential links between SAR produced by the wireless devices and cancers, the nervous system, vision and hearing, sleep, and effects on the growth of children; and (v) possible changes of DNA sequence that might cause the abnormal operation of bio-cells.

**Author Contributions:** M.A.-A., A.S.I.A., I.E., C.Z., N.O.P., A.U. and Y.A.-Y. jointly developed the first draft. R.L., C.H.S., D.Z., M.A. (Mohammad Alibakhshikenari), R.H. and Z.Z.A. revised the SAR standards and RF hazards. F.E., M.A. (Musa Abusitta) and F.M.A.A. presented the techniques utilised between 1978 and 2021. N.J.M., J.R., J.M.N. and R.A.A.-A. revised the final draft for submission. All authors have read and agreed to the published version of the manuscript.

**Funding:** The European Union's Horizon 2020 innovation programme under grant agreement H2020-MSCA-ITN-2016 SECRET-722424 and the U.K. Engineering and Physical Sciences Research Council (EPSRC) under grant EP/E022936/1 supported this research. This study is also sponsored by the FCT/MEC through national funds and, where applicable, co-financed by the ERDF under the PT2020 Partnership Agreement under the UID/EEA/50008/2020 project. This study is part of the POSITION-II project, which is funded by the European Commission's Joint Undertaking under the grant number Ecsel-7831132-Postitio-II-2017-IA. Moreover, this paper is also partially funded by British Council "2019 UK-China-BRI Countries Partnership Initiative" programme, with project titled "Adapting to Industry 4.0 oriented International Education and Research Collaboration". In addition, this project has received funding from Universidad Carlos III de Madrid and the European Union's Horizon 2020 research and innovation program under the Marie Sklodowska-Curie Grant 801538.

**Acknowledgments:** This paper acknowledges and is dedicated to the hard work of the long-standing colleague, Professor Peter S. Excell, from Bradford University and Wrexham University, U.K., who died in September 2020.

**Conflicts of Interest:** The authors declare no conflict of interest.

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
