# Peer review of "Wireless Electromagnetic Radiation Assessment Based on the Specific Absorption Rate (SAR): A Review Case Study"

_electronics, doi:10.3390/electronics11040511_

Round 1

Reviewer 1 Report

The manuscript is a good review ones on the topic of the radio frequency hazards due to the SAR. The various safety standards and organizations are included in the review. Also the published papers on the modelling and measurements of the SAR are summarized. The content of the review is interesting and organized well. Several comments on the review: 1.Necessary figures should be added for the reading. Long paragraphs of the description are too hard to follow. 2.A few of typos in the manuscript. Please make the correction.

Author Response

Reviewers’ comments are presented verbatim in italic black text.

Our response to each comment is presented in plain red text.

Reviewer 1:

The manuscript is a good review ones on the topic of the radio frequency hazards due to the SAR. The various safety standards and organizations are included in the review. Also the published papers on the modelling and measurements of the SAR are summarized. The content of the review is interesting and organized well.

Response: We would like to thank you for taking the time to carefully review our manuscript and for providing us with useful comments. We have revised the manuscript according to the suggestions of the reviewer and have responded to all of the comments. We do hope that these revisions are satisfactory.

Several comments on the review:

1.Necessary figures should be added for the reading. Long paragraphs of the description are too hard to follow.

2.A few of typos in the manuscript. Please make the correction.

Response: These are the valid comments. New Figures 2-6, 8-13 were added to enhance the explanation of texts. Some of the long paragraphs of the description have been shorted to make them easy to follow. Also, the typos were identified and corrected in the revised paper.

Reviewer 2 Report

This paper is a review of what is known about SARs (=Specific Absorption Rate) primarily for electromagnetic fields from cell phones. Since SARs is an important measure of concern to the health of cell phone users, it is considered of interest to have known results handily available. Here are some points to be  considered. 1. The results of this paper would have its main impact on the biomedical engineering field. So this journal may not be the best place for its publication, especially as there is little directly addressed to electronics. 2. If published here there should be more on the antennas of interest (including figures of them), the EM fields they produce (again with figures/mathematics) and how they contribute to SARs. 3. There are lots of abbreviations some of which are not defined, such as MoM and FDTD, and others at the place of first mention, as ICNIRP. 4. The same symbol should be used throughout for the same thing, as rho at lines 237, (?)252 and p at 637. 5. References for each of the lines in Table 1 would be useful (as they are in Table 2). 5. The program NEURON allows for the effect of the E field on a single neuron - that should be of interest to users of this paper. 6. The paper would also be very useful if the authors could point out what problems are left open especially for beginning researchers in the field. 7. The English though generally understandable does have a number of problems so it would best go through a certified technical English editor. 

Author Response

Reviewers’ comments are presented verbatim in italic black text.

Our response to each comment is presented in plain red text.

Reviewer 2:

This paper is a review of what is known about SARs (=Specific Absorption Rate) primarily for electromagnetic fields from cell phones. Since SARs is an important measure of concern to the health of cell phone users, it is considered of interest to have known results handily available. Here are some points to be considered. 1. The results of this paper would have its main impact on the biomedical engineering field. So this journal may not be the best place for its publication, especially as there is little directly addressed to electronics.

Response: Thank you for this interesting comment. As we mentioned in our “Introduction” section, the SAR is not only for cell phones. It is also will be used to measure the induced electromagnetic field on the IoT and wearable device users as the predicted number of IoT devices will reach 25.4 billion in 2030 [xx] from 8.74 billion in 2020. Hence, this paper plays a crucial role to increase the public awareness of the SARs.

[xx] https://www.statista.com/statistics/1183457/iot-connected-devices-worldwide/

  1. If published here there should be more on the antennas of interest (including figures of them), the EM fields they produce (again with figures/mathematics) and how they contribute to SARs.

Response: New Figures 2-6, 8-13 were added to enhance the explanation of this valid point.

  1. There are lots of abbreviations some of which are not defined, such as MoM and FDTD, and others at the place of first mention, as ICNIRP.

Response: The MoM, FDTD, ICNIRP were defined when they were first used in the revised manuscript.

  1. The same symbol should be used throughout for the same thing, as rho at lines 237, (?) 252 and p at 637.

Response: Thank you for your comments. This issue was corrected in the revised manuscript.

  1. References for each of the lines in Table 1 would be useful (as they are in Table 2).

Response: Thank you for raising this good point. We have revised this in our revised manuscript.

  1. The program NEURON allows for the effect of the E field on a single neuron - that should be of interest to users of this paper.

Response:

Thank you for this notification; we have added the following references in supporting the interaction between NEURON and E/H fields. These are: 

Ye H, Steiger A. Neuron matters: electric activation of neuronal tissue is dependent on the interaction between the neuron and the electric field. Journal of neuro-engineering and rehabilitation. 2015 Dec;12(1):1-9.

Boshuo Wang, Warren M. Grill, and Angel V. Peterchev, Coupling Magnetically Induced Electric Fields to Neurons: Longitudinal and Transverse Activation, Biophysical Journal 115, 95–107, July 3, 2018.

  1. The paper would also be very useful if the authors could point out what problems are left open especially for beginning researchers in the field.

Response:

This is a very good suggestion. Authors totally agreed with this. In the revised paper, section “Conclusion” has been modified to section “Conclusion and Future works”. The future direction of the research works was discussed in this section. Please see a new paragraph in the revised manuscript.

  1. The English though generally understandable does have a number of problems so it would best go through a certified technical English editor. 

Response: This paper was carefully proofread by a native English speaker.

Round 2

Reviewer 2 Report

This is a very nicely revised paper worthy of publication.